# One-loop amplitudes for $t\bar{t}j$ and $t\bar{t}\gamma$ productions at the LHC through $\mathcal{O}(\epsilon^2)$

**Souvik Bera[1]♣, Colomba Brancaccio[2]♡, Dhimiter Canko[3]♠ and Heribertus Bayu Hartanto[1,4]◇**

**1** Asia Pacific Center for Theoretical Physics, Pohang, 37673, Korea
**2** Dipartimento di Fisica and Arnold-Regge Center, Università di Torino, and INFN, Sezione di Torino, Via P. Giuria 1, I-10125 Torino, Italy
**3** Dipartimento di Fisica e Astronomia, Università di Bologna e INFN, Sezione di Bologna, via Irnerio 46, I-40126 Bologna, Italy
**4** Department of Physics, Pohang University of Science and Technology, Pohang, 37673, Korea

♣ souvik.bera@apctp.org , ♡ colomba.brancaccio@unito.it , ♠ dhimiter.canko2@unibo.it , ◇ bayu.hartanto@apctp.org

## Abstract

We present analytic expressions for the one-loop QCD helicity amplitudes contributing to top-quark pair production in association with a photon or a jet at the Large Hadron Collider (LHC), evaluated through $\mathcal{O}(\epsilon^2)$ in the dimensional regularisation parameter, $\epsilon$. These amplitudes are required to construct the two-loop hard functions that enter the NNLO QCD computation. The helicity amplitudes are expressed as linear combinations of algebraically independent components of the $\epsilon$-expanded master integrals, with the corresponding rational coefficients written in terms of momentum-twistor variables. We derive differential equations for the pentagon functions, which enable efficient numerical evaluation via generalised power series expansion method.

# 1 Introduction

The top-quark sector of the Standard Model of particle physics (SM) has been extensively studied at hadron collider experiments, providing stringent tests of the SM and offering a window into the exploration of Beyond the Standard Model (BSM) mechanisms [1–4]. In contrast to other strongly interacting particles in the SM, the top quark has the unique property of decaying before forming an hadronic bound state, a consequence of its large mass and short lifetime. As a result, its decay products not only provide direct access to the study of the top-quark properties, but also retain information through spin correlations, which persist because of the short decay time. Top-quark pair production in association with an additional particle, such as a jet ($t\bar{t}j$), a Higgs boson ($t\bar{t}H$), or a vector boson ($t\bar{t}V$, where $V \in \{\gamma, W^{\pm}, Z\}$), warrants a wide range of phenomenological investigations, due to the rich kinematics of $2 \to 3$ scattering processes.

In this article, we focus on processes involving top-quark pair production in association with a massless particle, specifically $pp \to t\bar{t}j$ and $pp \to t\bar{t}\gamma$. The $t\bar{t}\gamma$ production mechanism provides direct access to measure the top-quark charge [5] and allows to probe the structure of the top-photon vertex [6, 7] within the Standard Model Effective Field Theory (SMEFT) framework. In particular, it enables us to constraint anomalous electric dipole moments of the top quark, which are sensitive to physics beyond the SM [8–10]. On the other hand, $t\bar{t}j$ production at the LHC is key to study additional jet activity in $pp \to t\bar{t} + X$ events, since about half of these events are produced in association with extra jets [11–15]. In addition, it has been shown that, through $t\bar{t}j$ production, it is possible to extract the top-quark mass ($m_t$), as the normalized kinematic distribution constructed from the inverse $t\bar{t}j$ invariant mass is highly sensitive to $m_t$ variations [16–18]. Furthermore, both $t\bar{t}j$ and $t\bar{t}\gamma$ production processes are also instrumental for investigating the top-quark charge asymmetry [19], and they serve as important SM backgrounds in many searches for new physics at the LHC.

Theoretical predictions for $pp \to t\bar{t}j$ and $pp \to t\bar{t}\gamma$ processes are available up to next-to-leading order (NLO) accuracy, including both QCD and EW corrections [20–35]. With experimental measurements reaching unprecedented precision in current and upcoming LHC runs, theoretical predictions must be provided at least at NNLO in QCD in order to match the experimental precision. Achieving this level of accuracy for either the $t\bar{t}j$ or $t\bar{t}\gamma$ production is a highly non-trivial task due to the algebraic and analytic complexity arising in the computation of the two-loop $2 \to 3$ scattering amplitudes. Significant progress has recently been made in evaluating two-loop $2 \to 3$ QCD scattering amplitudes involving massless internal particles [36–50]. This progress has been made possible thanks to the introduction of finite-field methods [51, 52], which streamline computations by avoiding intermediate large expressions. These also include those arising from the reduction of Feynman integrals via integration-by-parts identities (IBPs) [53–55] into a minimal basis known as master integrals (MIs). In addition, the efficient computation of the MIs via the differential equations method

(DEs) [56–60], has been enabled by the use of the canonical form of DEs [61] and the construction of bases of functions consisting of the independent components of the $\epsilon$-expanded MIs, commonly known in the literature as *pentagon-functions* [62–64]. Building upon these developments, two-loop Feynman integral studies have been initiated for processes involving top quarks such as $t\bar{t}j$ [65–67], $t\bar{t}H$ [68] and $t\bar{t}W$ [69] productions, and numerical results for the two-loop amplitudes are now available for the leading colour $t\bar{t}j$ production in gluon fusion [70] and for the $N_f$ part of the process $q\bar{q} \to t\bar{t}H$ [71]. Moreover, in order to obtain NNLO QCD predictions, it is necessary to subtract both ultraviolet (UV) and infrared (IR) singularities from the two-loop scattering amplitudes and extract the finite remainder. This subtraction procedure requires UV renormalisation counterterms and the knowledge of the universal IR singularities, along with one-loop amplitudes expanded up to $\mathcal{O}(\epsilon^2)$ in the dimensional regularisation parameter, $\epsilon$. While several automated packages are available to perform a computation of one-loop amplitudes [72–76], they can only be used of a calculation up to the finite part, $\mathcal{O}(\epsilon^0)$, and the inclusion of higher $\epsilon$ terms will still require a dedicated effort. In this context, the relevant one-loop amplitudes for $2 \to 3$ processes involving a top-quark pair in the final state have been computed up to $\mathcal{O}(\epsilon^2)$ for the production channels: $gg/q\bar{q} \to t\bar{t}g$ [77], $gg \to t\bar{t}H$ [78] and $u\bar{d} \to t\bar{t}W^+$ [79].

The canonical DEs for the MIs associated to the pentagon topologies contributing to the one-loop $t\bar{t}j$ amplitude, as well as to the $t\bar{t}\gamma$ one-loop amplitude, have been studied in ref. [77], where the MIs have been evaluated by means of the generalised series expansion method [80], as implemented in DIFFEXP [81]. In this work, we take a further step by expressing the MIs in terms of pentagon functions. Expressing the amplitude in terms of pentagon functions provides a natural framework for performing the Laurent expansion in the dimensional regulator, which is essential for isolating UV and IR divergences and extracting the finite remainder analytically. Another benefit of using a pentagon-function basis is that it usually leads to a significant simplification of the final expressions. In order to perform numerical evaluations of the amplitudes, we construct DEs for the pentagon-function basis and solve them numerically via generalised power series expansions, utilizing the packages DIFFEXP and LINE [82]. We compute analytically the rational coefficients multiplying the pentagon functions using finite-field reconstruction techniques, which have been employed in numerous one- and two-loop multi-scale amplitude calculations. These techniques allow us to handle the complexity of the expressions at intermediate steps of the computation by replacing symbolic operations with numerical ones. We apply this method by exploiting the FINITEFLOW framework [52, 83]. Furthermore, we work in the helicity-amplitude representation, which provides a framework to efficiently include the top-quark decay process while preserving the information on spin correlation.

This paper is organized as follows. In section 2, we describe the colour structure of the one-loop amplitudes entering the $t\bar{t}j$ and $t\bar{t}\gamma$ production computations, as well as the helicity-amplitude computation starting from Feynman-diagram input. In section 3, we discuss the construction of the pentagon-function basis, the derivation of DEs for this basis and their numerical evaluation. In section 4, we provide numerical benchmark results of colour and helicity summed one-loop squared matrix elements up to $\mathcal{O}(\epsilon^2)$ for all possible channels appearing in the $pp \to t\bar{t}j$ and $pp \to t\bar{t}\gamma$ scattering processes. Finally, we conclude in section 5 with a summary of the results presented.

## 2 Helicity amplitudes

We compute one-loop QCD helicity amplitudes for the following partonic scattering processes entering the $pp \to t\bar{t}j$ production

$$0 \to \bar{t}(p_1) + t(p_2) + g(p_3) + g(p_4) + g(p_5), \tag{1}$$

$$0 \to \bar{t}(p_1) + t(p_2) + \bar{q}(p_3) + q(p_4) + g(p_5), \tag{2}$$

and the $pp \to t\bar{t}\gamma$ production

$$0 \to \bar{t}(p_1) + t(p_2) + g(p_3) + g(p_4) + \gamma(p_5), \tag{3}$$

$$0 \to \bar{t}(p_1) + t(p_2) + \bar{q}(p_3) + q(p_4) + \gamma(p_5), \tag{4}$$

where $q$ represents a generic massless quark species. Some representative Feynman diagrams contributing to these processes are depicted in fig. 1.

All the external momenta, $p_i$, which live in the 4-dimensional space-time, are taken to be outgoing

$$p_1 + p_2 + p_3 + p_4 + p_5 = 0, \tag{5}$$

and fulfil the following on-shell conditions

$$p_1^2 = p_2^2 = m_t^2 \qquad \text{and} \qquad p_3^2 = p_4^2 = p_5^2 = 0. \tag{6}$$

We choose the following six scalar products and one pseudo-scalar invariant to describe the kinematics of the processes in eqs. (1) to (4),

$$\vec{x} = \left( d_{12}, d_{23}, d_{34}, d_{45}, d_{15}, m_t^2 \right), \tag{7}$$

$$\text{tr}_5 = 4\mathrm{i}\,\varepsilon_{\mu\nu\rho\sigma}\, p_1^\mu p_2^\nu p_3^\rho p_4^\sigma, \tag{8}$$

where $d_{ij} = p_i \cdot p_j$ and $\varepsilon_{\mu\nu\rho\sigma}$ is the anti-symmetric Levi-Civita pseudo-tensor.

### 2.1 $t\bar{t}j$ partial amplitudes

The colour decomposition of the $L$-loop amplitude in terms of partial amplitudes for the $0 \to \bar{t}tggg$ scattering process is given by

$$\mathcal{M}^{(L)}\left(1_{\bar{t}}, 2_t, 3_g, 4_g, 5_g\right) = \sqrt{2}\, g_s^3 \left[ (4\pi)^\epsilon e^{-\epsilon\gamma_E} \frac{\alpha_s}{4\pi} \right]^L \times$$

$$\left\{ \sum_{\sigma \in S_3} (t^{a_{\sigma(3)}} t^{a_{\sigma(4)}} t^{a_{\sigma(5)}})_{i_2}^{\bar{i}_1} \, \mathcal{A}_{g;1}^{(L)}\left(1_{\bar{t}}, 2_t, \sigma(3)_g, \sigma(4)_g, \sigma(5)_g\right) \right.$$

$$+ \sum_{\sigma \in S_3'} \delta^{a_{\sigma(3)} a_{\sigma(4)}} (t^{a_{\sigma(5)}})_{i_2}^{\bar{i}_1} \, \mathcal{A}_{g;2}^{(L)}\left(1_{\bar{t}}, 2_t, \sigma(3)_g, \sigma(4)_g, \sigma(5)_g\right)$$

$$\left. + \sum_{\sigma \in \tilde{S}_3} \delta_{i_2}^{\bar{i}_1} \text{tr}(t^{a_{\sigma(3)}} t^{a_{\sigma(4)}} t^{a_{\sigma(5)}}) \, \mathcal{A}_{g;3}^{(L)}\left(1_{\bar{t}}, 2_t, \sigma(3)_g, \sigma(4)_g, \sigma(5)_g\right) \right\}, \tag{9}$$

where $g_s$ is the strong coupling constant, $t^a$ are the $SU(N_c)$ generators in the fundamental representation normalised according to $\text{tr}(t^a t^b) = \delta^{ab}/2$, $S_3' = S_3/\mathbb{Z}_2$, $\tilde{S}_3 = S_3/\mathbb{Z}_3$ and $\alpha_s = g_s^2/(4\pi)$. The sets of permutations entering eq. (9) are

$$S_3 = \{(3,4,5), (3,5,4), (4,3,5), (4,5,3), (5,3,4), (5,4,3)\},$$

$$S_3' = \{(3,4,5), (3,5,4), (4,5,3)\},$$

$$\tilde{S}_3 = \{(3,4,5), (3,5,4)\}.$$

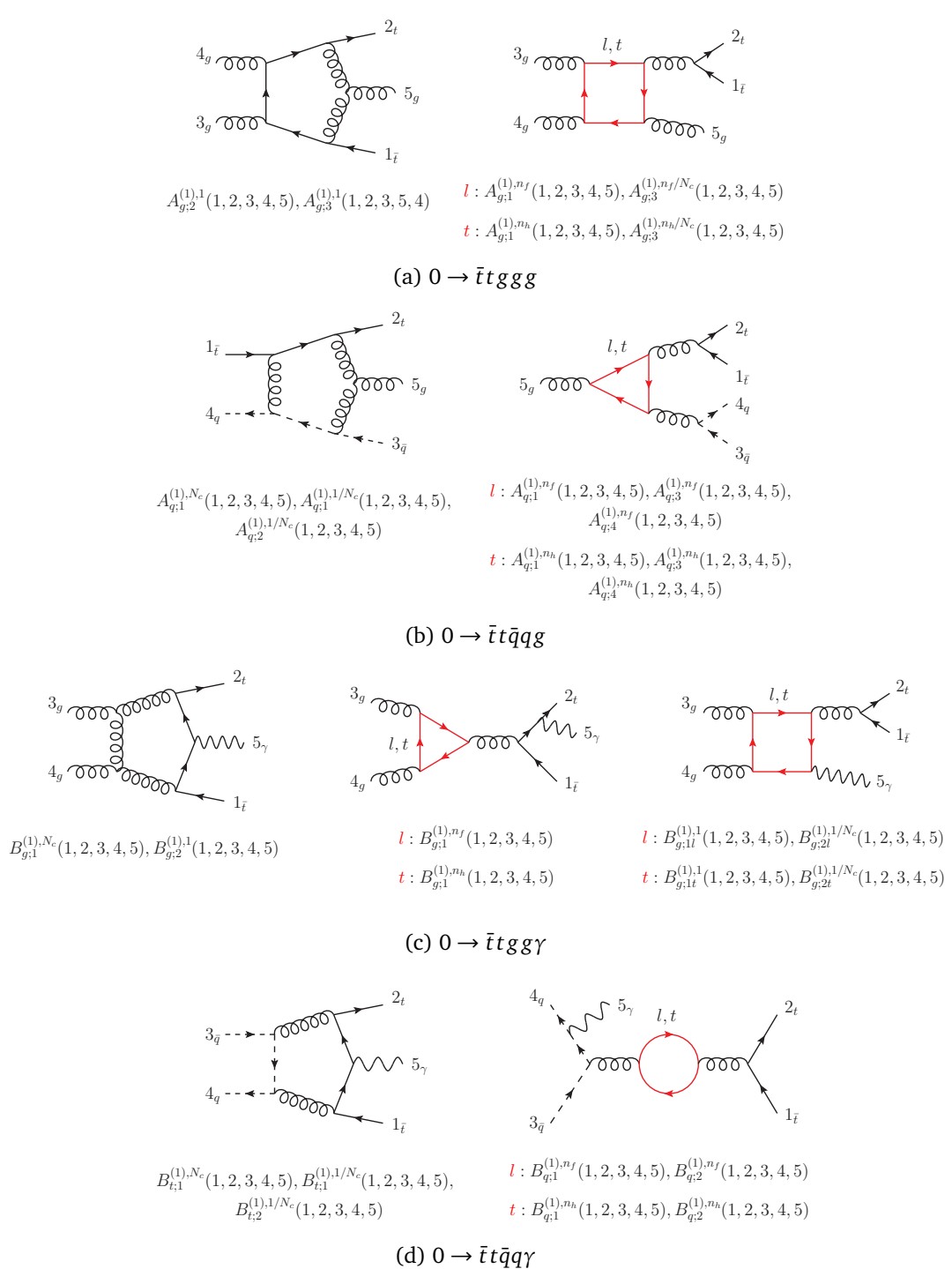

Figure 1: Sample Feynman diagrams contributing to one-loop $pp \to t\bar{t}j$ and $pp \to t\bar{t}\gamma$ amplitudes, together with the partial amplitudes that they contribute to. Black solid lines represent top quarks while black dashed lines represent massless external quarks. Moreover, wave lines denote photons, and curly lines denote gluons. Either massless quark ($l \in \{u, d, c, s, b\}$), or top-quark ($t$) can circulate in the red closed fermion loop, giving rise to $n_l$-, $n_h$-, $1l/2l$- and $1t/2t$-type partial amplitudes.

Each partial amplitude, $\mathcal{A}_{g;i}^{(L)}$, can further be decomposed according to their dependence on $N_c$ as well as the number of massless ($n_f$) and massive ($n_h$) closed fermion loops. At one-loop

level, this decomposition reads

$$\mathcal{A}_{g;1}^{(1)} = N_c A_{g;1}^{(1),N_c} + \frac{1}{N_c} A_{g;1}^{(1),1/N_c} + n_f A_{g;1}^{(1),n_f} + n_h A_{g;1}^{(1),n_h},\tag{10a}$$

$$\mathcal{A}_{g;2}^{(1)} = A_{g;2}^{(1),1},\tag{10b}$$

$$\mathcal{A}_{g;3}^{(1)} = A_{g;3}^{(1),1} + \frac{n_f}{N_c} A_{g;3}^{(1),n_f/N_c} + \frac{n_h}{N_c} A_{g;3}^{(1),n_h/N_c}.\tag{10c}$$

In the case of $0 \to \bar{t}t\bar{q}qg$ process, the $L$-loop colour decomposition is given by

$$\mathcal{M}^{(L)}\left(1_{\bar{t}}, 2_t, 3_{\bar{q}}, 4_q, 5_g\right) = \sqrt{2}\, g_s^3 \left[ (4\pi)^\epsilon e^{-\epsilon\gamma_E} \frac{\alpha_s}{4\pi} \right]^L \times$$
$$\left\{ \delta_{i_4}^{\bar{i}_1} (t^{a_5})_{i_2}^{\bar{i}_3}\ \mathcal{A}_{q;1}^{(L)}\left(1_{\bar{t}}, 2_t, 3_{\bar{q}}, 4_q, 5_g\right) \right.$$
$$+ \delta_{i_2}^{\bar{i}_3} (t^{a_5})_{i_4}^{\bar{i}_1}\ \mathcal{A}_{q;2}^{(L)}\left(1_{\bar{t}}, 2_t, 3_{\bar{q}}, 4_q, 5_g\right)\tag{11}$$
$$+ \frac{1}{N_c} \delta_{i_2}^{\bar{i}_1} (t^{a_5})_{i_4}^{\bar{i}_3}\ \mathcal{A}_{q;3}^{(L)}\left(1_{\bar{t}}, 2_t, 3_{\bar{q}}, 4_q, 5_g\right)$$
$$\left. + \frac{1}{N_c} \delta_{i_4}^{\bar{i}_3} (t^{a_5})_{i_2}^{\bar{i}_1}\ \mathcal{A}_{q;4}^{(L)}\left(1_{\bar{t}}, 2_t, 3_{\bar{q}}, 4_q, 5_g\right) \right\},$$

and the one-loop partial amplitude decomposition in terms of $N_c$, $n_f$ and $n_h$ reads

$$\mathcal{A}_{q;z}^{(1)} = N_c A_{q;z}^{(1),N_c} + \frac{1}{N_c} A_{q;z}^{(1),1/N_c} + n_f A_{q;z}^{(1),n_f} + n_h A_{q;z}^{(1),n_h},\tag{12}$$

where $z \in \{1, \ldots, 4\}$.

We derive analytic expressions for the gauge invariant mass-renormalised amplitude

$$\mathcal{A}_{\text{mren}}^{(1)} = \mathcal{A}^{(1)} + \delta Z_m^{(1)} \mathcal{A}_{\text{mct}}^{(0)},\tag{13}$$

which is obtained by adding the tree level amplitude with the one-loop mass counterterm insertion ($\mathcal{A}_{\text{mct}}^{(0)}$) to the bare amplitude ($\mathcal{A}^{(1)}$). Furthermore, $\delta Z_m^{(1)}$ is denoting the one-loop top-quark mass-renormalisation constant. The full UV-renormalised amplitude is obtained by including the counterterms associated with the strong coupling constant, as well as the renormalisation of the top-quark and gluon external wavefunctions,

$$\mathcal{A}_{\text{ren}}^{(1)} = \mathcal{A}_{\text{mren}}^{(1)} + \left( \delta Z_t^{(1)} + \frac{3}{2} \delta Z_{\alpha_s}^{(1)} + \frac{n_g}{2} \delta Z_g^{(1)} \right) \mathcal{A}^{(0)},\tag{14}$$

where $n_g$ is the number of external gluons involved in the scattering process ($n_g = 3$ for $0 \to \bar{t}tggg$ and $n_g = 1$ for $0 \to \bar{t}t\bar{q}qg$). The expressions of the one-loop renormalisation constants $\delta Z_m^{(1)}$, $\delta Z_t^{(1)}$ and $\delta Z_{\alpha_s}^{(1)}$ are provided in the appendix A.

## 2.2 $t\bar{t}\gamma$ partial amplitudes

We now turn to the colour decomposition of the $L$-loop amplitudes contributing to the production of $t\bar{t}\gamma$ final state in $pp$ collisions. For the gluon channel, $0 \to \bar{t}tgg\gamma$, the decomposition

is given by

$$
\mathcal{M}^{(L)}\left(1_{\bar{t}}, 2_t, 3_g, 4_g, 5_\gamma\right) = \sqrt{2}\, e\, g_s^2 \left[(4\pi)^\epsilon e^{-\epsilon\gamma_E} \frac{\alpha_s}{4\pi}\right]^L \times
$$

$$
\left\{ (t^{a_3} t^{a_4})_{i_2}^{\bar{i}_1}\, \mathcal{B}_{g;1}^{(L)}\left(1_{\bar{t}}, 2_t, 3_g, 4_g, 5_\gamma\right) \right.
$$

$$
+ (t^{a_4} t^{a_3})_{i_2}^{\bar{i}_1}\, \mathcal{B}_{g;1}^{(L)}\left(1_{\bar{t}}, 2_t, 4_g, 3_g, 5_\gamma\right)
$$

$$
\left. + \delta_{i_2}^{\bar{i}_1} \delta^{a_3 a_4}\, \mathcal{B}_{g;2}^{(L)}\left(1_{\bar{t}}, 2_t, 3_g, 4_g, 5_\gamma\right) \right\}, \tag{15}
$$

where $e$ is the magnitude of the electron charge. Similar to the $t\bar{t}j$ amplitudes, the $t\bar{t}\gamma$ one-loop partial amplitudes, $\mathcal{B}_{g;i}^{(L)}$, are further decomposed into sub-amplitudes according to the different types of closed fermion loops present:

$$
\mathcal{B}_{g;1}^{(1)} = Q_t N_c B_{g;1}^{(1),N_c} + \frac{Q_t}{N_c} B_{g;1}^{(1),1/N_c} + Q_t n_f B_{g;1}^{(1),n_f} + Q_t n_h B_{g;1}^{(1),n_h} + \tilde{Q}_l B_{g;1l}^{(1),1} + Q_t B_{g;1t}^{(1),1}, \tag{16a}
$$

$$
\mathcal{B}_{g;2}^{(1)} = B_{g;2}^{(1),1} + \tilde{Q}_l B_{g;2l}^{(1),1/N_c} + Q_t B_{g;2t}^{(1),1/N_c}. \tag{16b}
$$

In the expressions above, $\tilde{Q}_l = \sum_l Q_l$, with $Q_t$ and $Q_l$ being the fractional charges of the top-quark and of the massless quark ($l \in \{u, d, c, s, b\}$), respectively. In addition to the $n_f$- and $n_h$-type amplitudes, the $0 \to \bar{t}tgg\gamma$ colour decomposition contains contributions from diagrams where the photon couples to the internal quark closed-loop instead of the top-quark line ($B_{g;1l}^{(1),1}$, $B_{g;1t}^{(1),1}$, $B_{g;2l}^{(1),1/N_c}$, $B_{g;2t}^{(1),1/N_c}$), as shown in the rightmost Feynman diagram in fig. 1c.

For the $0 \to \bar{t}t\bar{q}q\gamma$ scattering amplitude, the $L$-loop colour decomposition is

$$
\mathcal{M}^{(L)}\left(1_{\bar{t}}, 2_t, 3_{\bar{q}}, 4_q, 5_\gamma\right) = \sqrt{2}\, e\, g_s^2 \left[(4\pi)^\epsilon e^{-\epsilon\gamma_E} \frac{\alpha_s}{4\pi}\right]^L \times
$$

$$
\left\{ \delta_{i_4}^{\bar{i}_1} \delta_{i_2}^{\bar{i}_3} \left[ Q_t\, \mathcal{B}_{t;1}^{(L)}\left(1_{\bar{t}}, 2_t, 3_{\bar{q}}, 4_q, 5_\gamma\right) + Q_q\, \mathcal{B}_{q;1}^{(L)}\left(1_{\bar{t}}, 2_t, 3_{\bar{q}}, 4_q, 5_\gamma\right) \right] \right.
$$

$$
\left. + \frac{1}{N_c} \delta_{i_2}^{\bar{i}_1} \delta_{i_4}^{\bar{i}_3} \left[ Q_t\, \mathcal{B}_{t;2}^{(L)}\left(1_{\bar{t}}, 2_t, 3_{\bar{q}}, 4_q, 5_\gamma\right) + Q_q\, \mathcal{B}_{q;2}^{(L)}\left(1_{\bar{t}}, 2_t, 3_{\bar{q}}, 4_q, 5_\gamma\right) \right] \right\}. \tag{17}
$$

In this case, the photon can be coupled either to the top quark or to the massless quark $q$, such that the corresponding partial amplitude picks up either $Q_t$ or the fractional charge of the massless quark $Q_q$. Equivalently to the $0 \to \bar{t}t\bar{q}qg$ case, the one-loop $0 \to \bar{t}t\bar{q}q\gamma$ partial amplitudes can be decomposed into sub-amplitude according to the $N_c$, $n_f$ and $n_h$ contribution,

$$
\mathcal{B}_{z;i}^{(1)} = N_c B_{z;i}^{(1),N_c} + \frac{1}{N_c} B_{z;i}^{(1),1/N_c} + n_f B_{z;i}^{(1),n_f} + n_h B_{z;i}^{(1),n_h}, \tag{18}
$$

where $z \in \{t, q\}$ and $i \in \{1, 2\}$. We note that in $0 \to \bar{t}t\bar{q}q\gamma$ process, partial amplitudes where the photon is coupled to the internal quark loop do not appear due to Furry's theorem.

The mass-renormalised amplitude, which is gauge invariant, is given by

$$
\mathcal{B}_{\mathrm{mren}}^{(1)} = \mathcal{B}^{(1)} + \delta Z_m^{(1)} \mathcal{B}_{\mathrm{mct}}^{(0)}, \tag{19}
$$

while the fully renormalised $t\bar{t}\gamma$ amplitude is constructed by including the appropriate UV counterterms,

$$
\mathcal{B}_{\mathrm{ren}}^{(1)} = \mathcal{B}_{\mathrm{mren}}^{(1)} + \left( \delta Z_t^{(1)} + \delta Z_{\alpha_s}^{(1)} + n_g \delta Z_g^{(1)} \right) \mathcal{B}^{(0)}. \tag{20}
$$

Here, we have $n_g = 2$ for $0 \to \bar{t}tgg\gamma$, and $n_g = 0$ for $0 \to \bar{t}t\bar{q}q\gamma$.

## 2.3  Helicity-amplitude computation

Our construction of helicity amplitudes for $pp \to t\bar{t}j$ and $pp \to t\bar{t}\gamma$ productions in the t'Hooft-Veltman scheme (tHV) starts with the decomposition of the partial amplitudes in terms of independent 4-dimensional tensor structures [84, 85]

$$
\begin{aligned}
A_{\mathrm{g}}^{(L)} &= \sum_{i=1}^{32} \mathcal{T}_{\mathrm{g},i} \, \mathcal{F}_{\mathrm{g},i}^{(L)} \,, \\
A_{\mathrm{q}}^{(L)} &= \sum_{i=1}^{16} \mathcal{T}_{\mathrm{q},i} \, \mathcal{F}_{\mathrm{q},i}^{(L)} \,,
\end{aligned}
\tag{21}
$$

where $A_{\mathrm{g}}^{(L)}$ represents either $0 \to \bar{t}tggg$ or $0 \to \bar{t}tgg\gamma$ amplitudes, while $A_{\mathrm{q}}^{(L)}$ stands for either $0 \to \bar{t}t\bar{q}qg$ or $0 \to \bar{t}t\bar{q}q\gamma$ amplitudes. The tensor structures for the $0 \to \bar{t}tggg$ and $0 \to \bar{t}tgg\gamma$ amplitudes are

$$
\begin{aligned}
\mathcal{T}_{\mathrm{g},1\ldots8} &= m_t^2 \, \bar{u}(p_2) v(p_1) \, \Gamma_{\mathrm{g},1\ldots8} \,, \\
\mathcal{T}_{\mathrm{g},9\ldots16} &= m_t \, \bar{u}(p_2) \slashed{p}_3 v(p_1) \, \Gamma_{\mathrm{g},1\ldots8} \,, \\
\mathcal{T}_{\mathrm{g},17\ldots24} &= m_t \, \bar{u}(p_2) \slashed{p}_4 v(p_1) \, \Gamma_{\mathrm{g},1\ldots8} \,, \\
\mathcal{T}_{\mathrm{g},25\ldots32} &= \bar{u}(p_2) \slashed{p}_3 \slashed{p}_4 v(p_1) \, \Gamma_{\mathrm{g},1\ldots8} \,,
\end{aligned}
\tag{22}
$$

while for the $0 \to \bar{t}t\bar{q}qg$ or $0 \to \bar{t}t\bar{q}q\gamma$ amplitudes they are

$$
\begin{aligned}
\mathcal{T}_{\mathrm{q},1\ldots4} &= m_t^2 \, \bar{u}(p_2) v(p_1) \, \Gamma_{\mathrm{q},1\ldots4} \,, \\
\mathcal{T}_{\mathrm{q},5\ldots8} &= m_t \, \bar{u}(p_2) \slashed{p}_3 v(p_1) \, \Gamma_{\mathrm{q},1\ldots4} \,, \\
\mathcal{T}_{\mathrm{q},9\ldots12} &= m_t \, \bar{u}(p_2) \slashed{p}_4 v(p_1) \, \Gamma_{\mathrm{q},1\ldots4} \,, \\
\mathcal{T}_{\mathrm{q},13\ldots16} &= \bar{u}(p_2) \slashed{p}_3 \slashed{p}_4 v(p_1) \, \Gamma_{\mathrm{q},1\ldots4} \,.
\end{aligned}
\tag{23}
$$

In eqs. (22) and (23) we have separated the top-quark spinor strings from the tensor structures of the massless particles. The tensor structures involving three massless vector bosons are given by

$$
\begin{aligned}
\Gamma_{\mathrm{g},1} &= p_1 \cdot \varepsilon(p_3) \, p_1 \cdot \varepsilon(p_4) \, p_1 \cdot \varepsilon(p_5) \,, \\
\Gamma_{\mathrm{g},2} &= p_1 \cdot \varepsilon(p_3) \, p_1 \cdot \varepsilon(p_4) \, p_2 \cdot \varepsilon(p_5) \,, \\
\Gamma_{\mathrm{g},3} &= p_1 \cdot \varepsilon(p_3) \, p_2 \cdot \varepsilon(p_4) \, p_1 \cdot \varepsilon(p_5) \,, \\
\Gamma_{\mathrm{g},4} &= p_2 \cdot \varepsilon(p_3) \, p_1 \cdot \varepsilon(p_4) \, p_1 \cdot \varepsilon(p_5) \,, \\
\Gamma_{\mathrm{g},5} &= p_1 \cdot \varepsilon(p_3) \, p_2 \cdot \varepsilon(p_4) \, p_2 \cdot \varepsilon(p_5) \,, \\
\Gamma_{\mathrm{g},6} &= p_2 \cdot \varepsilon(p_3) \, p_1 \cdot \varepsilon(p_4) \, p_2 \cdot \varepsilon(p_5) \,, \\
\Gamma_{\mathrm{g},7} &= p_2 \cdot \varepsilon(p_3) \, p_2 \cdot \varepsilon(p_4) \, p_1 \cdot \varepsilon(p_5) \,, \\
\Gamma_{\mathrm{g},8} &= p_2 \cdot \varepsilon(p_3) \, p_2 \cdot \varepsilon(p_4) \, p_2 \cdot \varepsilon(p_5) \,,
\end{aligned}
\tag{24}
$$

while for those involving a massless quark pair and a massless vector boson are

$$
\begin{aligned}
\Gamma_{\mathrm{q},1} &= \bar{u}(p_4) \slashed{p}_1 v(p_3) \, p_1 \cdot \varepsilon(p_5) \,, \\
\Gamma_{\mathrm{q},2} &= \bar{u}(p_4) \slashed{p}_1 v(p_3) \, p_2 \cdot \varepsilon(p_5) \,, \\
\Gamma_{\mathrm{q},3} &= \bar{u}(p_4) \slashed{p}_2 v(p_3) \, p_1 \cdot \varepsilon(p_5) \,, \\
\Gamma_{\mathrm{q},4} &= \bar{u}(p_4) \slashed{p}_2 v(p_3) \, p_2 \cdot \varepsilon(p_5) \,.
\end{aligned}
\tag{25}
$$

In writing $\Gamma_{\mathrm{g},i}$ and $\Gamma_{\mathrm{q},i}$, we have chosen $p_4$, $p_3$ and $p_3$ as the reference momenta of the polarisation vectors $\varepsilon(p_3)$, $\varepsilon(p_4)$ and $\varepsilon(p_5)$, respectively. The form factor, $\mathcal{F}_{\mathrm{g/q};i}^{(L)}$, is obtained by

multiplying the partial amplitudes in eq. (21) with the conjugated tensor structure $\mathcal{T}^{\dagger}_{\mathrm{g/q};i}$, summing over the polarisations of the external states, and inverting the resulting linear system of equations, so that

$$\mathcal{F}^{(L)}_{\mathrm{g/q};i} = \sum_{j=1}^{n_{\mathcal{T}}} \left(\Delta^{-1}_{\mathrm{g/q}}\right)_{ij} \tilde{A}^{(L)}_{\mathrm{g/q};j}, \tag{26}$$

where we have defined the so-called *contracted partial amplitude*

$$\tilde{A}^{(L)}_{\mathrm{g/q};i} = \sum_{\mathrm{pol}} T^{\dagger}_{\mathrm{g/q};i} A^{(L)}_{\mathrm{g/q}}, \tag{27}$$

while the linear system to be inverted is defined through the matrix

$$\Delta_{\mathrm{g/q};ij} = \sum_{\mathrm{pol}} \mathcal{T}^{\dagger}_{\mathrm{g/q};i} \mathcal{T}_{\mathrm{g/q};j}. \tag{28}$$

In eq. (26), $n_{\mathcal{T}}$ is the number of tensor structures appearing in eqs. (22) and (23), i.e. $n_{\mathcal{T}}(\mathrm{g}) = 32$ and $n_{\mathcal{T}}(\mathrm{q}) = 16$. We employ the following polarisation sum for the massless vector boson,

$$\sum_{\mathrm{pol}} \varepsilon^{*}_{\mu}(p_i, q_i) \varepsilon_{\nu}(p_i, q_i) = -g_{\mu\nu} + \frac{p_{i\mu}q_{i\nu} + q_{i\mu}p_{i\nu}}{p_i \cdot q_i}, \tag{29}$$

with $q$ being the massless reference momentum for the external polarisation vector.

The helicity amplitude is then obtained by specifying the helicity/spin states of the spinors and polarisation vectors in the tensor structures in eqs. (22) to (25). We follow the approach of ref. [86] to specify the spin states of the massive spinors, where the massive spinor is defined starting from the massless helicity spinor with the introduction of an arbitrary reference direction $n^{\mu}$

$$\bar{u}_{+}(p_2, m_t) = \frac{\langle n_2|(\not{p}_2 + m_t)}{\langle n_2 p_2^{\flat}\rangle}, \qquad v_{+}(p_1, m_t) = \frac{(\not{p}_1 - m_t)|n_1\rangle}{\langle p_1^{\flat} n_1\rangle}, \tag{30}$$

where we have used the following decomposition for the top quark and anti-top quark momenta ($p_1$ and $p_2$, respectively),

$$p_i^{\mu} = p_i^{\flat,\mu} + \frac{m_t^2}{p_i^{\flat} \cdot n_i} n_i^{\mu}, \tag{31}$$

with $n_i^2 = 0$, $(p_i^{\flat})^2 = 0$ and $i = 1, 2$. Having this massive spinor prescription, we can express the helicity amplitude in terms of massless momenta ($p_1^{\flat}, p_2^{\flat}, p_3, p_4, p_5$) in the basis of reference vectors $n_1$ and $n_2$ as follows [77, 87],

$$
\begin{aligned}
A^{(L)}(1_{\bar{t}}^{+}, 2_t^{+}, 3^{h_3}, 4^{h_4}, 5^{h_5}; n_1, n_2) = \frac{m_t \Phi^{h_3 h_4 h_5}}{\langle 1^{\flat} n_1\rangle \langle 2^{\flat} n_2\rangle} \Big\{ &\langle n_1 n_2\rangle s_{34} A^{(L),[1]}(1_{\bar{t}}^{+}, 2_t^{+}, 3^{h_3}, 4^{h_4}, 5^{h_5}) \\
&+ \langle n_1 3\rangle \langle n_2 4\rangle [34] A^{(L),[2]}(1_{\bar{t}}^{+}, 2_t^{+}, 3^{h_3}, 4^{h_4}, 5^{h_5}) \\
&+ \langle n_1 3\rangle \langle n_2 3\rangle \frac{[3|4|5|3]}{s_{34}} A^{(L),[3]}(1_{\bar{t}}^{+}, 2_t^{+}, 3^{h_3}, 4^{h_4}, 5^{h_5}) \\
&+ \langle n_1 4\rangle \langle n_2 4\rangle \frac{[4|5|3|4]}{s_{34}} A^{(L),[4]}(1_{\bar{t}}^{+}, 2_t^{+}, 3^{h_3}, 4^{h_4}, 5^{h_5}) \Big\}.
\end{aligned}
\tag{32}
$$

We only write down the "+" configuration for the massive spinor since the corresponding "-" configuration can be obtained by swapping the momenta $n \leftrightarrow p^{\flat}$ in eq. (30). In eq. (32), the

phase factors $\Phi$ depend on the helicity configuration of the massless particles. They are chosen in such a way that the sub-amplitudes $A^{(L),[i]}$ are free of spinor phase and dimensionless. For the $0 \to \bar{t}tggg$ and $0 \to \bar{t}tgg\gamma$ scattering processes, the phase factors $\Phi$ are

$$\Phi^{+++} = \frac{[35]}{s_{34}\langle 34 \rangle \langle 45 \rangle} , \tag{33a}$$

$$\Phi^{++-} = \frac{\langle 5|3|4|5 \rangle}{s_{34}^2 \langle 34 \rangle^2} , \tag{33b}$$

$$\Phi^{+-+} = \frac{\langle 4|5|3|4 \rangle}{s_{34}^2 \langle 35 \rangle^2} , \tag{33c}$$

$$\Phi^{-++} = \frac{\langle 3|5|4|3 \rangle}{s_{34}^2 \langle 45 \rangle^2} , \tag{33d}$$

while for the $0 \to \bar{t}t\bar{q}qg$ and $0 \to \bar{t}t\bar{q}q\gamma$, they are given by

$$\Phi^{+-+} = \frac{\langle 34 \rangle [35]}{s_{34}^2 \langle 35 \rangle} , \tag{34a}$$

$$\Phi^{-++} = \frac{\langle 34 \rangle [45]}{s_{34}^2 \langle 45 \rangle} . \tag{34b}$$

We computed the following helicity configurations for the massless particles: for the $0 \to \bar{t}tggg$ and $0 \to \bar{t}tgg\gamma$ processes, we considered the $+++$, $++-$, $+-+$, and $-++$ configurations. For the $0 \to \bar{t}t\bar{q}qg$ and $0 \to \bar{t}t\bar{q}q\gamma$ processes, we only considered the $+-+$ and $-++$ configurations. The remaining helicity configurations can be obtained by performing parity conjugation on the existing helicity amplitudes. We emphasize that our treatment of external massive spinor is not unique, and different representations can be employed [88], including the ones where a specific massive spinor description in not required, but instead using a form factor decomposotion only for massive spinor strings [78].

To derive the helicity sub-amplitudes ($A^{(L),[i]}$ in eq. (32)), we first evaluate the helicity amplitude in eq. (32) at four different choices of reference vectors

$$\mathfrak{A}^{(L),h_3 h_4 h_5} = \begin{pmatrix} A^{(L)}(1_{\bar{t}}^+, 2_t^+, 3^{h_3}, 4^{h_4}, 5^{h_5}; p_3, p_3) \\ A^{(L)}(1_{\bar{t}}^+, 2_t^+, 3^{h_3}, 4^{h_4}, 5^{h_5}; p_3, p_4) \\ A^{(L)}(1_{\bar{t}}^+, 2_t^+, 3^{h_3}, 4^{h_4}, 5^{h_5}; p_4, p_3) \\ A^{(L)}(1_{\bar{t}}^+, 2_t^+, 3^{h_3}, 4^{h_4}, 5^{h_5}; p_4, p_4) \end{pmatrix} , \tag{35}$$

which we will refer to as *projected helicity amplitudes* in the following. By inserting the four choices of $n_1$ and $n_2$ in eq. (32) and inverting the resulting system of equations, the helicity sub-amplitudes can be written in terms of projected helicity amplitude $\mathfrak{A}^{(L),h_3 h_4 h_5}$ elements as

$$A^{(L),[i]}(1_{\bar{t}}^+, 2_t^+, 3^{h_3}, 4^{h_4}, 5^{h_5}) = \sum_{j=1}^{4} \alpha_{ij}^{h_3 h_4 h_5} \, \mathfrak{A}_j^{(L),h_3 h_4 h_5} , \tag{36}$$

where the non-zero entries of $\alpha_{ij}^{h_3 h_4 h_5}$ are

$$
\begin{aligned}
\alpha_{12}^{h_3 h_4 h_5} = \alpha_{22}^{h_3 h_4 h_5} &= \frac{1}{m_t \Phi^{h_3 h_4 h_5}} \frac{\langle 1^\flat 3 \rangle \langle 2^\flat 4 \rangle}{s_{34} \langle 34 \rangle} \bigg|_{n_1 = p_3, n_2 = p_4}, \\
\alpha_{23}^{h_3 h_4 h_5} &= \frac{1}{m_t \Phi^{h_3 h_4 h_5}} \frac{\langle 1^\flat 4 \rangle \langle 2^\flat 3 \rangle}{s_{34} \langle 34 \rangle} \bigg|_{n_1 = p_4, n_2 = p_3}, \\
\alpha_{34}^{h_3 h_4 h_5} &= -\frac{1}{m_t \Phi^{h_3 h_4 h_5}} \frac{\langle 1^\flat 4 \rangle \langle 2^\flat 4 \rangle}{\langle 4|5|3|4 \rangle} \bigg|_{n_1 = p_4, n_2 = p_4}, \\
\alpha_{41}^{h_3 h_4 h_5} &= -\frac{1}{m_t \Phi^{h_3 h_4 h_5}} \frac{\langle 1^\flat 3 \rangle \langle 2^\flat 3 \rangle}{\langle 3|4|5|3 \rangle} \bigg|_{n_1 = p_3, n_2 = p_3}.
\end{aligned}
\tag{37}
$$

The projected helicity amplitudes $\mathfrak{A}^{(L), h_3 h_4 h_5}$ are obtained from the contracted partial amplitudes, $\tilde{A}_{g/q;i}^{(L)}$ in eq. (27), through

$$
\mathfrak{A}_i^{(L), h_3 h_4 h_5} = \sum_{j,k=1}^{n_{\mathcal{T}}} \Theta_{g/q;ij}^{h_3 h_4 h_5} \left( \Delta_{g/q}^{-1} \right)_{jk} \tilde{A}_{g/q;k}^{(L)}.
\tag{38}
$$

The helicity-dependent matrix $\Theta$ is built out of tensor structures in eqs. (22) and (23), where the spin/helicity states have been chosen such that

$$
\mathcal{T}_{g/q;i} \to \mathcal{T}_{g/q;i}^{++h_3 h_4 h_5}(n_1, n_2),
\tag{39}
$$

with the reference vectors $(n_1, n_2)$ evaluated at four different values as in eq. (35),

$$
\Theta_{g/q}^{h_3 h_4 h_5} = \begin{pmatrix}
\mathcal{T}_{g/q;1}^{++h_3 h_4 h_5}(p_3, p_3) & \cdots & \mathcal{T}_{g/q;n_{\mathcal{T}}}^{++h_3 h_4 h_5}(p_3, p_3) \\
\mathcal{T}_{g/q;1}^{++h_3 h_4 h_5}(p_3, p_4) & \cdots & \mathcal{T}_{g/q;n_{\mathcal{T}}}^{++h_3 h_4 h_5}(p_3, p_4) \\
\mathcal{T}_{g/q;1}^{++h_3 h_4 h_5}(p_4, p_3) & \cdots & \mathcal{T}_{g/q;n_{\mathcal{T}}}^{++h_3 h_4 h_5}(p_4, p_3) \\
\mathcal{T}_{g/q;1}^{++h_3 h_4 h_5}(p_4, p_4) & \cdots & \mathcal{T}_{g/q;n_{\mathcal{T}}}^{++h_3 h_4 h_5}(p_4, p_4)
\end{pmatrix}.
\tag{40}
$$

Finally, the helicity sub-amplitudes are derived from the contracted partial amplitudes $\tilde{A}_{g/q;i}^{(L)}$ as follows

$$
A^{(L),[i]}(1_{\bar{t}}^+, 2_t^+, 3^{h_3}, 4^{h_4}, 5^{h_5}) = \sum_{j=1}^{4} \sum_{k,l=1}^{n_{\mathcal{T}}} \alpha_{ij}^{h_3 h_4 h_5} \Theta_{g/q;jk}^{h_3 h_4 h_5} \left( \Delta_{g/q}^{-1} \right)_{kl} \tilde{A}_{g/q;l}^{(L)}.
\tag{41}
$$

Equipped with a prescription to construct helicity amplitudes for $pp \to t\bar{t}j$ and $pp \to t\bar{t}\gamma$, we now turn the discussion to the analytic computation framework. Our amplitude calculations follow the Feynman diagrammatic method, combined with finite-field arithmetic [51,52], which has been successfully used for the analytic computation of various two-loop five-point amplitudes (e.g., [37,42,49]). The calculation begins with the generation of Feynman diagrams using QGRAF [89], followed by the colour decomposition in order to extract partial amplitudes according to the structures discussed in section 2.1. For each partial amplitude, we construct the one-loop contracted partial amplitude $\tilde{A}_{g/q;i}^{(1)}$ using in-house scripts which utilise MATHEMATICA and FORM [90]. The one-loop contracted partial amplitudes are expressed as a linear combination of scalar Feynman integrals, which are further reduced onto the set of MIs introduced in ref. [77] by means of IBP reduction [53–55]. We make use of NEATIBP package [91] to generate an optimized system of IBP identities. The contracted partial amplitudes, as well as the MIs, are then Laurent-expanded around $\epsilon = 0$, keeping terms up to order $\epsilon^2$.

The components of the MIs in this expansion are written, order by order in $\epsilon$, as polynomials in the pentagon-function basis, the construction of which is discussed in detail in section 3.

The series of algebraic operations discussed so far are executed numerically over finite fields using the FINITEFLOW package [52, 83], starting with the contracted partial amplitude $\tilde{A}^{(1)}_{g/q;i}$, proceeding through the solution of the IBP identities, and ending with the determination of the one-loop helicity sub-amplitudes $A^{(1),[i]}$ as given in eq. (41). This framework allows us to evaluate numerically over finite fields the rational coefficients of the pentagon functions appearing in the helicity sub-amplitudes, which we exploit to reconstruct the corresponding analytic expressions. We employ several techniques to reduce the complexity of the analytical reconstruction, such as finding linear relations among rational coefficients, guessing the denominator of the rational coefficients and on-the-fly univariate partial fraction decomposition. For further details on the analytic reconstruction strategies that we used, see refs. [42, 44]. In addition, performing evaluations over finite fields requires a rational parametrisation of the external kinematics, for which we employ the momentum-twistor parametrisation [87, 92, 93] of ref. [77]. With this parametrisation, spinor products and Mandelstam invariants entering the $\alpha^{h_3 h_4 h_5}$, $\Theta^{h_3 h_4 h_5}_{g/q}$ and $\Delta_{g/q}$ matrices together with the contracted partial amplitude $\tilde{A}^{(L)}_{g/q}$ in eq. (41) can be written as rational functions in the momentum-twistor variables. The resulting analytic expressions for the mass-renormalised one-loop $pp \to t\bar{t}j$ and $pp \to t\bar{t}\gamma$ helicity sub-amplitudes are provided in the ancillary files accompanying this article [94] under the `anc/amplitudes/` directory.

## 3  Feynman integrals

In this section, we illustrate the Feynman integral families required for the computation of the one-loop $pp \to t\bar{t}\gamma$ and $pp \to t\bar{t}j$ scattering amplitudes. We employ the uniform transcendental basis of MIs introduced in ref. [66] to derive DEs [56–60] in canonical form [61]. We use the method of pentagon functions [37, 62–64, 68, 95], which has proven to be extremely useful in the computation of 2-loop 5-point massless amplitudes and recently for the computation of 2-loop 5-point amplitudes with massive internal propagators [70, 96]. The same approach was applied also for the computation up to $\mathcal{O}(\epsilon^2)$ of the 1-loop scattering amplitude of $t\bar{t}W$ production [79]. In line with these results, we express the one-loop $pp \to t\bar{t}\gamma$ and $pp \to t\bar{t}j$ scattering amplitudes up to $\mathcal{O}(\epsilon^2)$ in terms of pentagon functions, derive their DEs, and solve them numerically utilizing the generalized series expansion method [80].

In section 3.1, we introduce the notation and define the relevant integral families for the present calculation. In section 3.2, we briefly review the pentagon functions method, and in section 3.4, we discuss the numerical evaluation of the pentagon-function basis. This evaluation is carried out both using the MATHEMATICA package DIFFEXP [81] and the newly available C/C++ package LINE [82].

### 3.1  Definition of the integral families

The integral families required for the computation of the one-loop $pp \to t\bar{t}\gamma$ and $pp \to t\bar{t}j$ amplitudes can be collectively described by the formula

$$I^{\phi}_{n_1 n_2 n_3 n_4 n_5} = \int \frac{\mathrm{d}^d k_1}{i\pi^{d/2}} \frac{e^{\epsilon \gamma_E}}{D^{n_1}_{\phi,1} D^{n_2}_{\phi,2} D^{n_3}_{\phi,3} D^{n_4}_{\phi,4} D^{n_5}_{\phi,5}} \,, \tag{42}$$

where $d = 4-2\epsilon$ is the space-time dimension and $\phi \in \{P_A, P_B, P_C, P_D\}$. The inverse propagators $D_{\phi,i}$ are defined according to the conventions in ref. [66], and are listed in table 1 for the

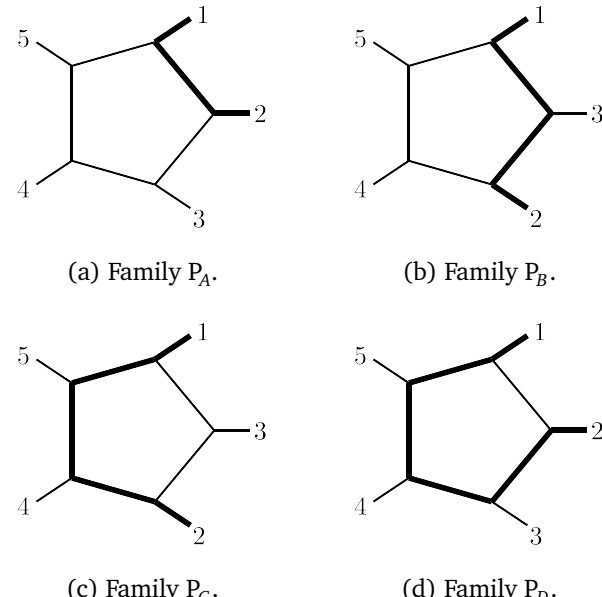

Figure 2: Graphical representation of the standard permutation of the four families contributing to the one-loop $pp \to t\bar{t}\gamma$ and $pp \to t\bar{t}j$ amplitudes. Thin black lines represent massless particles, while thick lines represent the top-quark.

standard permutation, $\sigma_0 = \{1, 2, 3, 4, 5\}$. We recall that labels 1 and 2 refer to the top and antitop quarks, respectively, while 3, 4, and 5 label the external massless particles. In fig. 2, each topology relevant to this computation is illustrated for the standard permutation, $\sigma_0$. However, to compute the squared amplitude, it is necessary to include contributions from the same topologies with the external gluons permuted. In particular, the relevant permutations are

$$\sigma \in \{(3, 4, 5), (3, 5, 4), (4, 3, 5), (4, 5, 3), (5, 3, 4), (5, 4, 3)\}, \tag{43}$$

where the fixed pair $(1, 2)$ is omitted for brevity.

|  | $P_A$ | $P_B$ | $P_C$ | $P_D$ |
|---|---|---|---|---|
| $D_{\phi,1}$ | $k_1^2$ | $k_1^2$ | $k_1^2 - m_t^2$ | $k_1^2 - m_t^2$ |
| $D_{\phi,2}$ | $(k_1 - p_1)^2 - m_t^2$ | $(k_1 - p_1)^2 - m_t^2$ | $(k_1 - p_1)^2$ | $(k_1 - p_1)^2$ |
| $D_{\phi,3}$ | $(k_1 - p_{12})^2$ | $(k_1 - p_{13})^2 - m_t^2$ | $(k_1 - p_{13})^2$ | $(k_1 - p_{12})^2 - m_t^2$ |
| $D_{\phi,4}$ | $(k_1 + p_{45})^2$ | $(k_1 + p_{45})^2$ | $(k_1 + p_{45})^2 - m_t^2$ | $(k_1 + p_{45})^2 - m_t^2$ |
| $D_{\phi,5}$ | $(k_1 + p_5)^2$ | $(k_1 + p_5)^2$ | $(k_1 + p_5)^2 - m_t^2$ | $(k_1 + p_5)^2 - m_t^2$ |

Table 1: Inverse propagators $D_{\phi,i}$ for the main permutation of the families contributing to the one-loop $pp \to t\bar{t}\gamma$ and $pp \to t\bar{t}j$ amplitudes. The shorthand notation $p_{i\ldots j} = p_i + \ldots + p_j$ is used for convenience.

By employing LITERED [97] for the generation of IBP identities [53–55] and FINITEFLOW [83] for solving them, we find for the integral families of type $P_A$, $P_B$, $P_C$, and $P_D$ a set of 15, 17, 19 and 21 MIs, respectively, in agreement with ref. [77]. Furthermore, by utilizing LITERED for obtaining mappings between the MIs of all families, including permutations, we identified a minimal set of 130 MIs, in terms of which the one-loop $pp \to t\bar{t}\gamma$ and $pp \to t\bar{t}j$ amplitudes can be expressed.

By using the bases of integrals provided in ref. [77], which are normalised to be free of $\epsilon$-poles, we construct canonical DEs for the standard permutations, $\sigma_0$, of all families, as well as for all the relevant permutations, $\sigma$. In particular, for the vector of uniform transcendental weight MIs of each family, $\vec{g}_{\phi,\sigma}(\vec{x})$, we obtain DEs of the form

$$\mathrm{d}\vec{g}_{\phi,\sigma}(\vec{x}) = \epsilon \, \mathrm{d}A_{\phi,\sigma}(\vec{x}) \, \vec{g}_{\phi,\sigma}(\vec{x}), \tag{44}$$

where the connection matrix, $\mathrm{d}A_{\phi,\sigma}(\vec{x})$, is written in dlog-form

$$\mathrm{d}A_{\phi,\sigma}(\vec{x}) = \sum_{i=1}^{283} a_{\phi,\sigma}^{(i)} \, \mathrm{dlog} \, W_i(\vec{x}). \tag{45}$$

We remind that $\vec{x}$ is the vector of kinematic invariants defined in eq. (7). In eq. (45), $a_{\phi,\sigma}^{(i)}(\vec{x})$ are matrices of rational numbers, $W_i(\vec{x})$ are algebraic functions of $\vec{x}$, known as letters, and the total set of letters is referred to as alphabet. For constructing the alphabet across all the families and their permutations, we collected all the letters and look for linear relations between them in order to identify a minimal set. We find 283 independent letters, of which 88 are rational functions of the kinematics, while the remaining ones are algebraic. The alphabet contains a total of 29 independent square roots, which we list in appendix B.

## 3.2 Pentagon-function basis

In amplitude computations, the $\epsilon$-pole structure is compared with the UV renormalisation and IR subtraction in order to obtain a consistency check as well as to extract the finite reminder. This requires Laurent expanding the scattering amplitude in $\epsilon$, which consequently involve Laurent expansion of the MIs contributing to it. In general, the components of the $\epsilon$-expansion of the MIs are not linearly independent, while it is desirable to express the amplitude in terms of a minimal set of independent components. The latter usually leads to significant simplifications in the final expression of the amplitude, to the analytic subtraction of $\epsilon$-poles, and improves the efficiency of the numerical evaluation. In this computation, we consider the components of the $\epsilon$-expansion of the MIs up to $\epsilon^4$, in order to retain all necessary information for future computations of the two-loop finite remainder for the processes addressed. We refer to the basis spanned by the independent components of the MIs as pentagon functions, reflecting the method's initial application to $2 \to 3$ scattering processes. However, we note that the procedure can be generalized to amplitudes with any number of external particles, e.g. [98–100].

To construct a pentagon-function basis related to a given set of MIs, one needs canonical DEs for the MIs and numerical boundary conditions for them. For the former, in section 3.1, we discussed how we obtained the canonical DEs for all the relevant families $\phi$ and permutations $\sigma$. For the latter, in order to determine the numerical boundary values of the MIs, we evaluate them at an arbitrary point of the physical phase space, $\vec{x}_0$, using AMFLOW [101] with 70-digit precision. The aforementioned MATHEMATICA package utilizes the auxiliary mass flow method [102] for the numerical evaluation of MIs.

With the above input, the procedure for creating the pentagon-function basis is straightforward. In the following, we provide only a sketch of the procedure; more comprehensive explanations can be found in refs. [37,62–64,79,95]. First, we Laurent expand the MIs around $\epsilon = 0$, so that

$$\vec{g}_{\phi,\sigma}(\vec{x},\epsilon) = \sum_{\omega=0}^{4} \epsilon^{\omega} \vec{g}_{\phi,\sigma}^{(\omega)}(\vec{x}) + \mathcal{O}(\epsilon^5), \tag{46}$$

where $\omega$ is the order of the expansion, $\vec{g}_{\phi,\sigma}^{(\omega)}(\vec{x})$ the components (of the expansion) of the MIs, and terms of order higher than $\epsilon^4$ are neglected. Then, we iteratively integrate the canonical DEs order by order in $\epsilon$ along a path $\gamma$ connecting the initial point $\vec{x}_0$ to an arbitrary point of the phase space $\vec{x}$. The solution of the DEs can be written in terms of Chen iterated integrals [103],

$$\vec{g}_{\phi,\sigma}^{(\omega)}(\vec{x}) = \sum_{\omega'=0}^{\omega} \sum_{i_1,i_2,\ldots,i_{\omega'}=1}^{283} a_{\phi,\sigma}^{(i_{\omega'})} a_{\phi,\sigma}^{(i_{\omega'-1})} \ldots a_{\phi,\sigma}^{(i_1)} \vec{g}_{\phi,\sigma}^{(\omega-\omega')}(\vec{x}_0) \left[ W_{i_1}, W_{i_2}, \ldots, W_{i_{\omega'}} \right]_{\vec{x}_0}(\vec{x}), \quad (47)$$

where

$$\left[ W_{i_1}, W_{i_2}, \ldots, W_{i_{\omega}} \right]_{\vec{x}_0}(\vec{x}) \equiv \int_0^1 dt\, \frac{\partial \log W_{i_{\omega}}(\gamma(t))}{\partial t} \left[ W_{i_1}, W_{i_2}, \ldots, W_{i_{\omega-1}} \right]_{\vec{x}_0}(\gamma(t)). \quad (48)$$

A very useful property of Chen iterated integrals is that they form a shuffle algebra. This means that the product of an iterated integral of weight $\omega_1$ with an iterated integral of weight $\omega_2$ can be expressed as a sum of iterated integrals of weight $\omega_1 + \omega_2$ according to the equation

$$\left[ W_{a_1}, \ldots, W_{a_{\omega_1}} \right]_{\vec{x}_0}(\vec{x}) \left[ W_{b_1}, \ldots, W_{b_{\omega_2}} \right]_{\vec{x}_0}(\vec{x}) = \sum_{\vec{c} = \vec{a} \sqcup \vec{b}} \left[ W_{c_1}, \ldots, W_{c_{\omega_1+\omega_2}} \right]_{\vec{x}_0}(\vec{x}), \quad (49)$$

where $\vec{a}, \vec{b}, \vec{c}$ are index vectors, and $\vec{a} \sqcup \vec{b}$ denotes all possible ways to shuffle the indices of $\vec{a}$ and $\vec{b}$ while preserving the order within each vector.

We proceed by identifying the relations among MIs at the symbol level [104–106]. These relations can then be lifted to the iterated integrals based on the ansatz, proposed in ref. [95], that only terms proportional to $\zeta$ need to be included. This procedure will be briefly reviewed later in the section. For now, we start by finding relations at the symbol level for the 130 independent MIs, denoted by $\vec{g}$. The MIs at symbol level can be written as

$$\mathcal{S}\left[ \vec{g}^{(\omega)} \right] = \sum_{i_1,\ldots,i_\omega} a^{(i_{\omega'})} a^{(i_{\omega'-1})} \ldots a^{(i_1)} \vec{g}^{(w-w')}(\vec{x}_0) W_{i_1} \otimes W_{i_2} \otimes \ldots \otimes W_{i_{\omega'}}. \quad (50)$$

Next, we identify the independent components of the MIs, weight by weight. At weight zero, we have only rational numbers, which can be written in terms of a single constant that we define as $F^{(0)} = 1$, where $F^{(0)}$ represents the pentagon function at weight zero. At weight one, we can derive the $\mathcal{Q}$-linear relations through the condition

$$\sum_{i=1}^{130} c_i \, \mathcal{S}\left[ g_i^{(1)} \right] = 0. \quad (51)$$

The procedure for higher weights is similar but not identical. More specifically, for weights $\omega \geq 2$, we must also consider, in the ansatz of eq. (51), products of lower-weight pentagon functions, which take the form

$$F_{i_1}^{(\omega_1)} F_{i_2}^{(\omega_2)} \ldots F_{i_k}^{(\omega_k)}, \quad (52)$$

with $\omega_1 + \omega_2 + \ldots + \omega_k = \omega$ and $1 \leq \omega_i < \omega$ for $i = 1, \ldots, k$ and $1 < k < \omega$. The symbol of the latter can be expressed in terms of weight $\omega$ symbols by exploiting shuffle algebra, eq. (49).

Subsequent to this step, the relations between the symbols of the ansatz can be found in the same manner as was done for the weight one case. Since the choice of independent MI components forming the pentagon function basis is arbitrary, we tune the ordering so that products of lower-weight pentagon functions are preferred over higher-weight ones. This helps minimize the number of higher-weight pentagon functions that appear in the basis. The number of independent pentagon functions at each weight is summarized in table 2.

| Weight | 0 | 1 | 2 | 3 | 4 | all |
|---|---|---|---|---|---|---|
| pentagon functions | 1 | 15 | 53 | 111 | 101 | 281 |

Table 2: Number of pentagon functions at each weight used in the computation of the full colour one-loop scattering amplitude up to $\mathcal{O}(\epsilon^2)$ for the processes $pp \to t\bar{t}\gamma$ and $pp \to t\bar{t}j$.

The last step consists of expressing the components of the MIs in terms of the constructed basis of pentagon functions. To do so, products of lower weight pentagon functions and transcendental constants with vanishing symbols need to be added in the ansatz defining the MI components in terms of pentagon functions. Therefore $\vec{g}^{(k)}(\vec{x})$ are expressed as polynomials in the basis of pentagon functions and the transcendental constants $i\pi$, $\zeta_2$, $\zeta_3$ and $\zeta_4$, with rational numbers as coefficients, as proposed in ref. [95]. For example, at weight two, we can make the ansatz

$$\vec{g}^{(2)}(\vec{x}) = \sum_i c^i F_i^{(2)} + \sum_{i,j} c^{i,j} F_i^{(1)} F_j^{(1)} + \sum_i \tilde{c}^i \, i\pi F_i^{(1)} + \tilde{c}\, \zeta_2 \,, \tag{53}$$

where the rational numbers $c^i$ and $c^{i,j}$ are already known from the symbol-level analysis. The coefficients $\tilde{c}$ and $\tilde{c}^i$ can be determined by ensuring consistency with the expression of the MI components in terms of Chen iterated integrals, as outlined in eq. (47). More specifically, eq. (53) is set equal to eq. (47) at weight-two, where the pentagon functions are expressed in terms of MI components and subsequently in terms of Chen iterated integrals. By utilizing the linear independence of the iterated integrals, we obtain a system of linear equations consisting of the boundary values $\{\vec{g}^{(0)}(\vec{x}_0), \vec{g}^{(1)}(\vec{x}_0), \vec{g}^{(2)}(\vec{x}_0)\}$ and the undetermined coefficients. Given that the boundary values are already computed using AMFLOW, we can determine the coefficients $\tilde{c}$ and $\tilde{c}^i$.

Proceeding in a similar way at each weight, we express all the MI components as polynomials with rational coefficients in the algebraically independent MI components, denoted as $\{F_i^{(k)}\}$ and known as the pentagon function basis, including also the transcendental constants $\{\zeta_2, \zeta_3, \zeta_4\}$. For example, the weight-two component of MI 36 is written as

$$\vec{g}_{36}^{(2)}(\vec{x}) = -\zeta_2 + 4F_1^{(1)} F_5^{(1)} - F_3^{(1)} F_5^{(1)} - 3\left(F_5^{(1)}\right)^2 + F_5^{(1)} F_7^{(1)} + 2F_5^{(1)} F_8^{(1)} \tag{54}$$
$$+ 2F_5^{(1)} F_{11}^{(1)} - 2F_5^{(1)} F_{13}^{(1)} + F_{20}^{(2)}$$

The analytic expressions of the MI components in terms of the pentagon functions are reported in the ancillary file `anc/specialfunc/SFbasis/MIs_to_SFs.m` [94].

## 3.3 Physical region

We focus on the $s_{34}$ scattering channel ($34 \to 125$), since the other channels relevant to the production processes $pp \to t\bar{t}\gamma$ and $pp \to t\bar{t}j$ can be derived by performing an appropriate permutation of the external momenta. We will describe how to handle these permutations in a next section. In the physical region, the momenta are required to be real and thus associated with physical scattering angles and positive energies. For the $s_{34}$ channel, this translates into the following conditions

$$p_1^2 > 0, \quad p_2^2 > 0, \quad p_3 \cdot p_4 > 0, \quad p_1 \cdot p_2 > 0, \quad p_1 \cdot p_5 > 0, \quad p_2 \cdot p_5 > 0,$$
$$p_1 \cdot p_3 < 0, \quad p_2 \cdot p_3 < 0, \quad p_3 \cdot p_5 < 0, \quad p_1 \cdot p_4 < 0, \quad p_2 \cdot p_4 < 0, \quad p_4 \cdot p_5 < 0, \tag{55}$$

Additionally, the following Gram determinant constraints must be satisfied [107,108]

$$\det G(p_i, p_j) < 0, \qquad \det G(p_i, p_j, p_k) > 0, \qquad \det G(p_i, p_j, p_k, p_l) < 0, \tag{56}$$

where $\{i, j, k, l\} \in \{1, \ldots, 5\}$. We refer to appendix B for the definition of the Gram determinant. These conditions, together with the previously stated constraints, impose a set of polynomial relations on the kinematic invariants that define the boundary of our physical region. These constrains can be found in the ancillary file `anc/specialfunc/DiffExpRun/s34_-channel.m` [94]. We comment that for the square roots with negative argument, we choose to have positive imaginary part.

### 3.4 Numerical evaluation of the pentagon functions

To enable analytic cancellation of the poles in the finite remainder, we have derived analytic expressions in terms of logarithms for the weight-one pentagon functions, in order to express the IR subtraction and UV factorisation terms in the same basis of functions as the amplitude. The analytic expressions are the following

$$F_1^{(1)} = \log\left(\frac{2(d_{12} + d_{23} - d_{45} + m_t^2)}{m_t^2}\right),$$

$$F_2^{(1)} = \frac{\log(m_t^2)}{2} + \log\left(\frac{d_{15} - d_{23} + d_{45}}{d_{34} + d_{45} - d_{12} - m_t^2}\right) - 2\log\left(2(d_{34} - d_{12} - d_{15} - m_t^2)\right) + 2i\pi,$$

$$F_3^{(1)} = \frac{\log(m_t^2)}{2} + \log\left(\frac{d_{34} - d_{12} - d_{15} - m_t^2}{d_{34}}\right) - 2\log\left(2(d_{15} - d_{23} + d_{45})\right),$$

$$F_4^{(1)} = \frac{\log(m_t^2)}{2} - \log\left(\frac{d_{34} - d_{12} - d_{15} - m_t^2}{d_{34}}\right) - \log\left(2(d_{12} + m_t^2)\right) + i\pi,$$

$$F_5^{(1)} = \log\left(2(d_{34} - d_{12} - d_{15} - m_t^2)\right) - \log(m_t^2) - i\pi,$$

$$F_6^{(1)} = \frac{\log(m_t^2)}{2} + \log\left(\frac{d_{12} + d_{23} - d_{45} + m_t^2}{d_{45}}\right) - 2\log\left(2(d_{34} - d_{12} - d_{15} - m_t^2)\right) + 2i\pi,$$

$$F_7^{(1)} = \frac{\log(m_t^2)}{2} + \log\left(\frac{d_{34} - d_{12} - d_{15} - m_t^2}{d_{34}}\right) - 2\log\left(2(d_{12} + d_{23} - d_{45} + m_t^2)\right),$$

$$F_8^{(1)} = \frac{\log(m_t^2)}{2} + \log\left(\frac{d_{23} + d_{34} - d_{15}}{d_{12} + d_{23} - d_{45} + m_t^2}\right) - \log\left(2d_{15}\right) + i\pi, \tag{57}$$

$$F_9^{(1)} = -\log\left(\frac{2(d_{12} + d_{23} - d_{45} + m_t^2)}{m_t^2}\right) - \log\left(-2d_{23}\right),$$

$$F_{10}^{(1)} = \log\left(-\frac{1 + r_{24}}{1 - r_{24}}\right),$$

$$F_{11}^{(1)} = \log(m_t^2) - \log\left(2(d_{23} + d_{34} - d_{15})\right) - \log\left(2(d_{15} - d_{23} + d_{45})\right),$$

$$F_{12}^{(1)} = \log\left(-\frac{1 + r_{25}}{1 - r_{25}}\right),$$

$$F_{13}^{(1)} = \log(m_t^2) - \log\left(2(d_{34} - d_{12} - d_{15} - m_t^2)\right) - \log\left(2d_{15}\right) + 2i\pi,$$

$$F_{14}^{(1)} = \log\left(-\frac{1 + r_{27}}{1 - r_{27}}\right) - i\pi,$$

$$F_{15}^{(1)} = -\log\left(-\frac{1 + r_4}{1 - r_4}\right) + i\pi.$$

where all the logarithms are well defined in the $s_{34}$ channel.

We do not derive explicit expressions for pentagon functions of higher weights. Instead, we provide tailored procedures for their numerical evaluation. To achieve this, we construct

a system of linear DEs for evaluating the basis of pentagon functions [37]. The DEs take the form

$$\mathrm{d}\,\vec{G}(\vec{x}) = \tilde{A}(\vec{x})\,\vec{G}(\vec{x})\,, \tag{58}$$

with the connection matrix having the standard dlog-form

$$\tilde{A}(\vec{x}) = \sum_i \tilde{a}_i \,\mathrm{dlog}\,W_i(\vec{x})\,. \tag{59}$$

Here, $\vec{G}(\vec{x})$ represents the minimal set of pentagon functions necessary to construct the linear DEs.

Let us now illustrate the procedure of finding the minimal basis $\vec{G}(\vec{x})$ and creating the aforementioned DEs. We begin by considering the weight-four pentagon functions, denoted as $\{F_i^{(4)}\}$. We compute their derivatives with respect to the kinematic invariants, leveraging the knowledge of DEs for the MIs. Indeed, the pentagon functions $\{F_i^{(4)}\}$ are by definition MIs components, i.e. $\vec{g}^{(4)}(\vec{x})$, whose derivatives are given by the $\epsilon$-expansion of eq. (44). The derivatives of the weight-four functions will only involve weight-three functions, including products of lower-weight functions whose total weight is equal to three. To obtain a closed system of DEs, we include these products in the set $\vec{G}(\vec{x})$, along with the weight-three pentagon functions $\{F_i^{(3)}\}$, and differentiate them accordingly. To ensure that we solve the DEs for a minimal set of functions $\vec{G}(\vec{x})$, we restrict the set of functions to only the independent linear combinations of lower weight products that appear in the DEs. This process is repeated at each weight until we reach weight zero. Upon completion, we obtain DEs for a total of 325 functions, which represents an extension of the original basis of pentagon functions $\{F_i^{(w)}\}$. The set of pentagon functions $\vec{G}(\vec{x})$ together with their DEs can be found in the ancillary file `anc/specialfunc/SFBasis/SF_DEs.m` [94].

The boundary values for the set of pentagon functions $\vec{G}(\vec{x})$ can be determined by using the numerical boundaries obtained for the MIs with AMFLOW. We provide the values of $\vec{G}(\vec{x})$ in four different phase-space points in the ancillary files `anc/specialfunc/DiffExpRun/Bounds/AMF_-X#.m` [94], with $\# = 1, 2, 3, 4$. These points are chosen to lie in the $s_{34}$ channel and are selected to avoid spurious singularities in DEs. The boundary values provided have been used to solve the DEs. The propagation of the solution from one of these benchmark points to another using the generalized series expansion [80], as implemented in DIFFEXP [81], served as a cross-check of the DEs. In `anc/specialfunc/DiffExpRun/DiffExp_ttA.wl` file [94] one can find our DIFFEXP setup. This setup can be used for obtaining solutions of the DEs for any target point connected to the boundary point through a path which entirely lies within the $s_{34}$ channel.

Additionally to the DIFFEXP implementation for the solution of the pentagon-function DEs, we provide also an implementation in terms of the newly available LINE package [82]. Since LINE can only handle connection matrices with rational functions, it is necessary to rotate the DEs to eliminate square roots from the connection matrices. We perform this rotation using the transformation rule

$$\vec{G}'(\vec{x}) = N\,\vec{G}(\vec{x}) \qquad \text{and} \qquad \tilde{A}'(\vec{x}) = N\,\tilde{A}(\vec{x})\,N^{-1} + \mathrm{d}N\,N^{-1}\,, \tag{60}$$

where the normalisation matrix $N$ is chosen such that the connection matrix $\tilde{A}'(\vec{x})$ is square-root free. LINE is used for solving the DEs of $\vec{G}'(\vec{x})$ and the solution of $\vec{G}(\vec{x})$ is obtained by using the inverse transformation. A script to rotate the DEs and prepare inputs for running LINE is provided in the ancillary file `anc/specialfunc/LINErun/LINEinput_ttA.wl` [94].

We emphasize that the basis for the pentagon functions has been constructed to include all permutations of the three external massless particles in eqs. (1) to (4). This ensures that

all colour configurations, shown in eqs. (9), (11), (15) and (17), as well as all helicity configurations necessary for evaluating the squared amplitudes of all the channels relevant for $t\bar{t}j$ and $t\bar{t}\gamma$ production in $pp$ collisions are covered. Thanks to this construction, once the DEs are solved, there is no need to re-evaluate the pentagon functions or perform any analytic continuation when permuting external legs to obtain other helicity or colour configurations beyond the independent ones. Everything is already embedded in the solution. To illustrate this, consider the evaluation of the pentagon function $F_2^{(1)}(\vec{x})$ at a phase-space point corresponding to the permutation

$$\sigma' : (p_1, p_2, p_3, p_4, p_5) \to (p_1, p_2, p_3, p_5, p_4). \tag{61}$$

We can derive the relation

$$F_2^{(1)}(\sigma' \cdot \vec{x}) = \sigma' \cdot F_2^{(1)}(\vec{x}) = \sigma' \cdot g_{2,\phi,\sigma}^{(1)}(\vec{x}) = g_{2,\phi,\sigma'\cdot\sigma}^{(1)}(\vec{x}) = F_7^{(1)}(\vec{x}) - 2F_8^{(1)}(\vec{x}) + F_{13}^{(1)}(\vec{x}). \tag{62}$$

This means that the evaluation of $F_2^{(1)}(\vec{x})$ at the permuted kinematic point can be obtained from a specific linear combination of pentagon functions evaluated at the original (non-permuted) point, which we choose to lie in the $s_{34}$ channel. In the file `anc/specialfunc/SFbasis/SFs_-perm_rule.m` [94], we provide a list of permutation rules for the pentagon functions.

   We further carried out performance studies for numerical evaluations using LINE and DIFFEXP. The timings were measured on 100 physical phase-space points,[1] and we found that, on average, LINE is about twice as fast as DIFFEXP in propagating solutions of the DEs while aiming for the same number of digits of accuracy in the results. Beyond speed, an additional advantage of LINE over DIFFEXP for large-scale cluster evaluations is that it does not require a MATHEMATICA license. Additionally, we also compared numerical evaluations based on the generalized series expansion method between the master-integral and pentagon-function representations. The DEs system for the MIs is significantly smaller, since it involves only a single permutation, whereas the pentagon functions cover the full set of permutations. However, for a given phase-space point, the numerical solutions for the MIs have to be computed multiple times (once for each permutation), while in the pentagon-function representation they are evaluated only once. We tested this using DIFFEXP on a few physical phase-space points and found the timings to be comparable between the two approaches. We note, however, that the computational advantage of the pentagon-function representation, in terms of evaluation speed, typically arises when employing one-fold integral representations [63, 64], which we leave for future work.

## 4   Numerical benchmark results

In this section, we employ the analytic expressions of the helicity amplitudes, together with the special function basis to obtain numerical results for colour- and helicity-summed UV renormalised squared amplitudes. For $pp \to t\bar{t}j$ production, we consider the following partonic scattering channels for the $0 \to \bar{t}tggg$ process,

$$gg \to \bar{t}tg: \quad g(-p_3) + g(-p_4) \to \bar{t}(p_1) + t(p_2) + g(p_5), \tag{63}$$

---

[1]We restricted to points that can be connected to the boundary point by a path that does not cross singularities. In the current version of LINE, crossing singularities requires manual tuning of the working precision for each run by trial and error, which we chose not to pursue in this analysis.

and for the $0 \to \bar{t}t\bar{q}qg$ process we have

$$u\bar{u} \to \bar{t}tg: \quad u(-p_3) + \bar{u}(-p_4) \to \bar{t}(p_1) + t(p_2) + g(p_5), \tag{64a}$$

$$\bar{u}u \to \bar{t}tg: \quad \bar{u}(-p_3) + u(-p_4) \to \bar{t}(p_1) + t(p_2) + g(p_5), \tag{64b}$$

$$ug \to \bar{t}tu: \quad u(-p_3) + g(-p_4) \to \bar{t}(p_1) + t(p_2) + u(p_5), \tag{64c}$$

$$\bar{u}g \to \bar{t}t\bar{u}: \quad \bar{u}(-p_3) + g(-p_4) \to \bar{t}(p_1) + t(p_2) + \bar{u}(p_5), \tag{64d}$$

$$gu \to \bar{t}tu: \quad g(-p_3) + u(-p_4) \to \bar{t}(p_1) + t(p_2) + u(p_5), \tag{64e}$$

$$g\bar{u} \to \bar{t}t\bar{u}: \quad g(-p_3) + \bar{u}(-p_4) \to \bar{t}(p_1) + t(p_2) + \bar{u}(p_5). \tag{64f}$$

For $pp \to t\bar{t}\gamma$ production, there is one partonic channel for $0 \to \bar{t}tgg\gamma$ process,

$$gg \to \bar{t}t\gamma: \quad g(-p_3) + g(-p_4) \to \bar{t}(p_1) + t(p_2) + \gamma(p_5), \tag{65}$$

while there are four channels in the $0 \to \bar{t}t\bar{q}q\gamma$ process,

$$u\bar{u} \to \bar{t}t\gamma: \quad u(-p_3) + \bar{u}(-p_4) \to \bar{t}(p_1) + t(p_2) + \gamma(p_5), \tag{66a}$$

$$d\bar{d} \to \bar{t}t\gamma: \quad d(-p_3) + \bar{d}(-p_4) \to \bar{t}(p_1) + t(p_2) + \gamma(p_5), \tag{66b}$$

$$\bar{u}u \to \bar{t}t\gamma: \quad \bar{u}(-p_3) + u(-p_4) \to \bar{t}(p_1) + t(p_2) + \gamma(p_5), \tag{66c}$$

$$\bar{d}d \to \bar{t}t\gamma: \quad \bar{d}(-p_3) + d(-p_4) \to \bar{t}(p_1) + t(p_2) + \gamma(p_5). \tag{66d}$$

To present a benchmark numerical evaluation, we employ the following physical phase-space point (given in the units of GeV) that lies in the $34 \to 125$ channel,

$$
\begin{aligned}
p_1 &= (421.812558294, 72.3820876039, 342.337070428, 163.070530566), \\
p_2 &= (463.866064984, -151.62473585, -322.765175691, -243.113773948), \\
p_3 &= (-500, 0, 0, -500), \\
p_4 &= (-500, 0, 0, 500), \\
p_5 &= (114.321376722, 79.2426482461, -19.5718947366, 80.0432433819),
\end{aligned}
\tag{67}
$$

which corresponds to the value of top-quark mass of $m_t = 170$ GeV, together with the following coupling constants and renormalisation scale,

$$\alpha_s = 0.118, \qquad \alpha = \frac{1}{137}, \qquad \mu_R = 1 \text{ GeV}. \tag{68}$$

In tables 3 and 4, we present numerical results for the the colour- and helicity-summed one-loop amplitude interefered with the tree-level amplitude, normalised with respect to the tree-level amplitude as

$$\frac{2\text{Re} \sum \mathcal{M}^{(0)*} \mathcal{M}^{(1)}}{\frac{\alpha_s}{4\pi} \sum |\mathcal{M}^{(0)}|^2}, \tag{69}$$

evaluated at the phase-space point given in eq. (67). These results have been validated against OPENLOOPS [76] up to $\mathcal{O}(\epsilon^0)$.

## 5 Conclusions

In this work, we have derived analytic expressions for the full-colour one-loop helicity amplitudes for $t\bar{t}\gamma$ and $t\bar{t}j$ production processes up to $\mathcal{O}(\epsilon^2)$. The motivation behind our computation lies in the growing need for precise scattering amplitudes results, through NNLO accuracy in the perturbative expansion around $\alpha_s = 0$. While several automated tools are available for

| $0 \to \bar{t}tggg$ | $\epsilon^{-2}$ | $\epsilon^{-1}$ | $\epsilon^0$ | $\epsilon^1$ | $\epsilon^2$ |
|---|---|---|---|---|---|
| $gg \to \bar{t}tg$ | $-18.000000$ | $196.49906$ | $-1289.8912$ | $4852.8064$ | $-13717.358$ |

| $0 \to \bar{t}t\bar{q}qg$ | $\epsilon^{-2}$ | $\epsilon^{-1}$ | $\epsilon^0$ | $\epsilon^1$ | $\epsilon^2$ |
|---|---|---|---|---|---|
| $u\bar{u} \to \bar{t}tg$ | $-11.333333$ | $119.80952$ | $-872.01453$ | $3501.7374$ | $-11272.732$ |
| $\bar{u}u \to \bar{t}tg$ | $-11.333333$ | $114.02910$ | $-798.36940$ | $3030.3552$ | $-9278.0196$ |
| $ug \to \bar{t}tu$ | $-11.333333$ | $134.05316$ | $-986.23890$ | $3843.0879$ | $-11171.395$ |
| $\bar{u}g \to \bar{t}t\bar{u}$ | $-11.333333$ | $132.56334$ | $-973.29477$ | $3791.4615$ | $-11047.051$ |
| $gu \to \bar{t}tu$ | $-11.333333$ | $121.80354$ | $-877.87570$ | $3399.8392$ | $-10211.249$ |
| $g\bar{u} \to \bar{t}t\bar{u}$ | $-11.333333$ | $129.14005$ | $-962.56681$ | $3882.2416$ | $-12038.663$ |

Table 3: Numerical results for $pp \to t\bar{t}j$ UV-renormalised one-loop amplitudes interfered with the corresponding tree level amplitudes, normalised according to eq. (69), evaluated at the phase-space point specified in eq. (67).

| $0 \to \bar{t}tgg\gamma$ | $\epsilon^{-2}$ | $\epsilon^{-1}$ | $\epsilon^0$ | $\epsilon^1$ | $\epsilon^2$ |
|---|---|---|---|---|---|
| $gg \to \bar{t}t\gamma$ | $-12.000000$ | $149.51298$ | $-1070.6693$ | $4402.2091$ | $-13362.539$ |

| $0 \to \bar{t}t\bar{q}q\gamma$ | $\epsilon^{-2}$ | $\epsilon^{-1}$ | $\epsilon^0$ | $\epsilon^1$ | $\epsilon^2$ |
|---|---|---|---|---|---|
| $u\bar{u} \to \bar{t}t\gamma$ | $-5.3333333$ | $71.531343$ | $-631.45963$ | $2904.3524$ | $-10280.372$ |
| $d\bar{d} \to \bar{t}t\gamma$ | $-5.3333333$ | $71.531343$ | $-633.66653$ | $2931.1602$ | $-10440.782$ |
| $\bar{u}u \to \bar{t}t\gamma$ | $-5.3333333$ | $64.932872$ | $-548.72683$ | $2389.0454$ | $-8147.6420$ |
| $\bar{d}d \to \bar{t}t\gamma$ | $-5.3333333$ | $64.932872$ | $-549.43852$ | $2391.3058$ | $-8150.1573$ |

Table 4: Numerical results for $pp \to t\bar{t}\gamma$ UV-renormalised one-loop amplitudes interfered with the corresponding tree level amplitudes, normalised according to eq. (69), evaluated at the phase-space point specified in eq. (67).

computing the finite part of one-loop amplitudes, two-loop computations require the one-loop expressions expanded up to $\mathcal{O}(\epsilon^2)$. Our results thus contribute to this goal, while at the same time offering insight into the algebraic complexity of the two-loop scattering amplitudes of $t\bar{t}j$ and $t\bar{t}\gamma$ production.

Within our calculation, we employed a tensor decomposition of the amplitudes in four dimensions [84, 85] to derive the helicity-amplitude representation. The algebraic complexity arising from the enormous size of intermediate expressions was managed using finite-field methods [51, 52, 83]. In order to express the amplitudes in a compact manner, we constructed a basis of pentagon functions [62–64], which is evaluated using generalised series expansion techniques [80]. In this context, we made use of the DIFFEXP package [81] and the newly released LINE package [82], which showed improved performances for our system of DEs. Our results represent a novel contribution for the one-loop scattering amplitude of $t\bar{t}\gamma$ production, while for the corresponding $t\bar{t}j$ case, the derivation of the pentagon function basis marks an advancement over previous computations [77].

The choice of expressing the amplitudes in terms of a pentagon-function basis is motivated not only by the resulting compactness and numerical efficiency of the one-loop expressions but also by the long-term goal of facilitating future two-loop computations of these processes. Previous analyses of leading-colour two-loop amplitudes for $t\bar{t}j$ production have shown that the relevant functional space extends beyond polylogarithmic functions and includes elliptic structures [65–67, 70]. As shown in the $t\bar{t}j$ case, although a complete set of independent pentagon functions was not yet available in the general context of elliptic functions, it was possible to construct a potentially overcomplete pentagon functions-like basis. This approach led to several advantages, including the separation of polylogarithmic and non-polylogarithmic contributions, with the latter pushed into the finite part of the amplitude, enabling analytic cancellation of poles and simplification of the amplitude expressions. Furthermore, the construction of this basis improved both numerical stability and efficiency. Given that the same Feynman integral topologies, along with more complex ones, are expected to appear in the two-loop leading-colour $t\bar{t}\gamma$ amplitude, we anticipate that a similar strategy could be effective in this case as well. This, however, remains the subject of future research.

# Acknowledgements

We thank Simon Badger, Matteo Bechetti, Tiziano Peraro, Mattia Pozzoli and Simone Zoia for fruitful discussions and comments on the draft. We are grateful to Mattia Pozzoli for kindly providing access to private code, which was crucial for the construction of pentagon functions basis. Thanks to Renato Maria Prisco for crucial discussions enabling us to provide numerical solution of the DEs within the LINE framework. S.B. and H.B.H. has been supported by an appointment to the JRG Program at the APCTP through the Science and Technology Promotion Fund and Lottery Fund of the Korean Government and by the Korean Local Governments – Gyeongsangbuk-do Province and Pohang City. C.B. acknowledges funding from the Italian Ministry of Universities and Research (MUR) through FARE grant R207777C4R. D.C. acknowledges support from the European Research Council (ERC) under the European Union's Horizon Europe research and innovation program grant agreement 101040760, *High-precision multi-leg Higgs and top physics with finite fields* (ERC Starting Grant FFHiggsTop).

# A   Renormalisation constants

The required one-loop QCD renormalisation constants are given by [109, 110]

$$\delta Z_m^{(1)} = Z_t^{(1)} = \frac{e^{\epsilon \gamma_E} \Gamma(1+\epsilon)}{(m_t^2)^\epsilon} C_F \left( -\frac{3}{\epsilon} - \frac{4}{1-2\epsilon} \right) , \tag{70a}$$

$$\delta Z_g^{(1)} = \frac{e^{\epsilon \gamma_E} \Gamma(1+\epsilon)}{(m_t^2)^\epsilon} T_F n_h \left( -\frac{4}{3\epsilon} \right) , \tag{70b}$$

$$\delta Z_{\alpha_s}^{(1)} = -\frac{\beta_0}{\epsilon} + \frac{4}{3} T_F n_h \left[ \frac{1}{\epsilon} - \log(m_t^2) \right] , \tag{70c}$$

where for the strong coupling renormalisation counterterm, the $n_f$ light quarks are subtracted using the $\overline{\text{MS}}$ scheme while the heavy quark loop is subtracted at zero momentum. The one-loop $\beta$-function coefficient is

$$\beta_0 = \frac{11}{3} C_A - \frac{4}{3} T_F n_f , \tag{71}$$

with

$$C_F = \frac{N_c^2 - 1}{2 N_c} , \qquad C_A = N_c \qquad \text{and} \qquad T_F = \frac{1}{2} . \tag{72}$$

# B  Independent square roots

In this appendix, we list the independent square roots appearing in the computation of the one-loop $pp \to t\bar{t}\gamma$ and $pp \to t\bar{t}j$ squared amplitudes. The list of square roots is:

$$r_1 = \sqrt{\text{tr}_5^2/4}\,, \qquad\qquad r_2 = \sqrt{-\det G(p_2, p_1 + p_5)}\,,$$

$$r_3 = \sqrt{-\det G(p_1, p_2 + p_3)}\,, \qquad\qquad r_4 = \sqrt{\frac{d_{12} - m_t^2}{d_{12} + m_t^2}}\,,$$

$$r_5 = \sqrt{-\det G(p_2, p_1 + p_4)}\,, \qquad\qquad r_6 = \sqrt{-\det G(p_1, p_2 + p_5)}\,,$$

$$r_7 = \sqrt{\frac{\det Y^{0 m_t m_t 0}_{p_1|p_5|p_2|p_3+p_4}}{d_{25}^2 d_{15}^2}}\,, \qquad\qquad r_8 = \sqrt{\frac{\det Y^{m_t 0 m_t m_t}_{p_1+p_3|p_2|p_4|p_5}}{d_{24}^2 d_{45}^2}}\,,$$

$$r_9 = \sqrt{\frac{\det Y^{m_t 0 m_t m_t}_{p_1|p_2+p_3|p_4|p_5}}{d_{15}^2 d_{45}^2}}\,, \qquad\qquad r_{10} = \sqrt{-\det G(p_2, p_1 + p_3)}\,,$$

$$r_{11} = \sqrt{\frac{\det Y^{m_t 0 m_t m_t}_{p_1+p_3|p_2|p_5|p_4}}{d_{25}^2 d_{45}^2}}\,, \qquad\qquad r_{12} = \sqrt{\frac{\det Y^{m_t 0 m_t m_t}_{p_1|p_2+p_3|p_5|p_4}}{d_{14}^2 d_{45}^2}}\,,$$

$$r_{13} = \sqrt{\frac{\det Y^{0 m_t m_t m_t}_{p_2|p_3|p_5|p_1+p_4}}{d_{23}^2 d_{35}^2}}\,, \qquad\qquad r_{14} = \sqrt{\frac{\det Y^{m_t 0 m_t m_t}_{p_1|p_2+p_4|p_3|p_5}}{d_{15}^2 d_{35}^2}}\,,$$

$$r_{15} = \sqrt{-\det G(p_1, p_2 + p_4)}\,, \qquad\qquad r_{16} = \sqrt{\frac{\det Y^{m_t 0 m_t m_t}_{p_1+p_4|p_2|p_5|p_3}}{d_{25}^2 d_{35}^2}}\,, \qquad (73)$$

$$r_{17} = \sqrt{\frac{\det Y^{m_t 0 m_t m_t}_{p_1|p_2+p_4|p_5|p_3}}{d_{13}^2 d_{35}^2}}\,, \qquad\qquad r_{18} = \sqrt{\frac{\det Y^{0 m_t m_t m_t}_{p_2|p_3|p_4|p_1+p_5}}{d_{23}^2 d_{34}^2}}\,,$$

$$r_{19} = \sqrt{\frac{\det Y^{0 m_t m_t m_t}_{p_2|p_4|p_3|p_1+p_5}}{d_{24}^2 d_{34}^2}}\,, \qquad\qquad r_{20} = \sqrt{\frac{\det Y^{m_t 0 m_t m_t}_{p_1|p_2+p_5|p_3|p_4}}{d_{14}^2 d_{34}^2}}\,,$$

$$r_{21} = \sqrt{\frac{\det Y^{m_t 0 m_t m_t}_{p_1|p_2+p_5|p_4|p_3}}{d_{13}^2 d_{34}^2}}\,, \qquad\qquad r_{22} = \sqrt{\frac{\det Y^{0 m_t m_t 0}_{p_1|p_3|p_2|p_4+p_5}}{d_{23}^2 d_{13}^2}}\,,$$

$$r_{23} = \sqrt{\frac{\det Y^{0 m_t m_t 0}_{p_1|p_4|p_2|p_3+p_5}}{d_{24}^2 d_{14}^2}}\,, \qquad\qquad r_{24} = \sqrt{1 - \frac{2 m_t^2}{d_{45}}}\,,$$

$$r_{25} = \sqrt{1 - \frac{2 m_t^2}{d_{12} - d_{34} - d_{45} + m_t^2}}\,, \qquad\qquad r_{26} = \sqrt{\frac{\det Y^{m_t m_t m_t m_t}_{p_1+p_2|p_3|p_4|p_5}}{d_{34}^2 d_{45}^2}}\,,$$

$$r_{27} = \sqrt{1 - \frac{2 m_t^2}{d_{34}}}\,, \qquad\qquad r_{28} = \sqrt{\frac{\det Y^{m_t m_t m_t m_t}_{p_1+p_2|p_3|p_5|p_4}}{d_{35}^2 d_{45}^2}}\,,$$

$$r_{29} = \sqrt{\frac{\det Y^{m_t m_t m_t m_t}_{p_1+p_2|p_4|p_5|p_3}}{d_{34}^2 d_{35}^2}}\,,$$

with the Gram and Cayley matrices are defined by

$$G(p_1, \cdots, p_n) = \begin{pmatrix} p_1 \cdot p_1 & p_1 \cdot p_2 & \cdots & p_1 \cdot p_n \\ p_2 \cdot p_1 & p_2 \cdot p_2 & \cdots & p_2 \cdot p_n \\ \vdots & \vdots & \ddots & \vdots \\ p_n \cdot p_1 & p_n \cdot p_2 & \cdots & p_n \cdot p_n \end{pmatrix}, \qquad \left[ Y_{P_1|P_2|P_3|P_4}^{m_1 m_2 m_3 m_4} \right]_{ij} = \frac{1}{2} \left[ (q_i - q_j)^2 - m_i^2 - m_j^2 \right],$$

(74)

where $q_i = \sum_{k=0}^{i-1} P_k$.

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
