# Peer review of "One-loop amplitudes for $\ttj$ and $\ttgamma$ productions at the LHC through $\mathcal{O}(\epsilon^2)$"

_SciPost Physics, doi:SciPost Phys. 19, 165 (2025)_

## Round 1 · Referee Report · Anonymous (Referee 1) · 2025-10-14

Strengths

1-Phenomenological relevance for the LHC.
2-Timeliness.

Weaknesses

1-The presentation is unclear in several points.
2-Lack of relevant details.

Report

The authors discuss the analytic computation of the full-color one-loop scattering amplitudes for two processes: the production of a pair of top quarks in association with a jet ($t\bar{t}j$) and with a photon ($t\bar{t}\gamma$) at hadron colliders. These processes are of significant relevance for the LHC and are currently attracting considerable attention, particularly in efforts to compute NNLO QCD corrections. Computing the required two-loop amplitudes is the main bottleneck, owing to the emergence of elliptic functions in the Feynman integrals, paired with the complex five-particle kinematics.

One-loop amplitudes for these processes are already implemented in automated tools such as OpenLoops, and NLO predictions have been available for some time. However, the purpose of this article is not to provide new phenomenologically usable one-loop results, but rather to probe the complexity and prepare the ground for the challenging two-loop computation. For this reason, the Laurent expansion around $\epsilon = 0$, where $\epsilon$ is the dimensional regulator, is truncated at order $\epsilon^2$ to supply the terms needed for the subtraction of poles at two loops. Similar studies have been published for $t\bar{t}$ production in association with a Higgs and a $W$ boson.

Owing to the phenomenological relevance of these processes, the growing attention they are receiving, and the complexity of the two-loop computations, I believe this article would satisfy SciPost’s acceptance criteria, provided that the authors address the points of criticism listed in the attachment.

Requested changes

The list of requested changes is rather lengthy, hence I give it in the attachment for better formatting.

Attachment

Recommendation

Ask for minor revision

  • validity: good
  • significance: good
  • originality: ok
  • clarity: ok
  • formatting: excellent
  • grammar: good

Author:  Dhimiter Canko  on 2025-11-14  [id 6031]

(in reply to Report 1 on 2025-10-14)

We would like to thank both referees for their thoughtful and constructive comments and suggestions. We believe their feedback has been very helpful and has significantly improved the quality of our paper. Here we answer the comments and questions raised by Referee #1.

We will address the major and minor points raised by Referee #1 in detail in this response. We do not repeat referee's comments/questions in this response for conciseness.

1) We follow the referee's suggestion by extracting the finite remainder of the one-loop amplitude for ttj and ttgamma processes. In particular, we recast the analytic expressions provided in the ancillary files from previously just the mass-renormalised representation, now delivered in three different parts: the finite remainders, ultra-violet poles and infra-red poles. From these 3 ingredients one can build either mass-renormalised or fully renormalised amplitude. The Mathematica script to evaluate the colour- and spin-summed squared matrix element has also been updated. We discuss this finite remainder extraction and the new structure of ancillary files in the second-to-last paragraph of Section 2.

2) We agree that while the complexity of the rational functions of the two-loop amplitude is typically orders of magnitude higher than that of O(eps^2) term at one-loop, we can still learn some valuable information from the observation made in the one-loop amplitude computation. Firstly, we provide the maximum polynomial degrees in the numerator (as a proxy to the complexity of rational coefficients of the pentagon function), for the most complicated partial amplitudes in both 0->ttggg and 0->ttgga processes, showing the comparison between leading and sub-leading colour contributions. We also highlight the effectiveness of the univariate partial fraction decomposition method to reduce the complexity of performing analytic reconstruction from finite-field evaluation. In addition we made a comment about performing multivariate partial fraction decomposition on our analytic expressions, which unfortunately results in larger analytic expressions. We elaborate our observations on the algebraic complexity of the one-loop analytic expressions in the last paragraph of Section 2.

3) The referee correctly noted that further clarification would be helpful regarding the restriction of the numerical evaluation routine to paths lying within the s34s_{34}s34 channel, as well as the extent of the reachable phase space. In response, we have expanded the discussion in the revised manuscript to better explain these aspects. The updated text now reads:

“This setup can be employed to evaluate the solution of the DEs for any target point connected to a boundary point by a path that lies entirely within the $s_{34}$ channel. The integration path may, in principle, extend beyond the physical region; however, this would require crossing branch cuts associated with physical singularities. In such cases, an appropriate analytic continuation prescription must be supplied to the DE solver. To circumvent this complication, for target points not directly connected to the available boundary conditions within the physical region, an intermediate point connected to both a boundary point and the target can be introduced. The special functions evaluated at this intermediate point then serve as boundary conditions for propagating the DEs solution to the target point. Furthermore, we observe that starting from the four phase-space points at which the special functions are provided in the ancillary files, our numerical implementation provides coverage of the entire physical region. This test has been perfomed for $100 k$ phase-space points generated with \texttt{MadGraph5}~\cite{Alwall:2014hca} for $pp \to t \bar{t} \gamma$ at LO.”

4) The second to last sentence in the abstract has been modified as follows:

"The helicity amplitudes are expressed as linear combinations of algebraically independent components of the $\eps$-expanded master integrals, with the corresponding rational coefficients ... "

---->

"The helicity amplitudes are expressed as linear combinations of algebraically independent components of the $\eps$-expanded master integrals--known as pentagon function--with the corresponding rational coefficients ... "

5) We rewrite the final sentence of the abstract from

"We derive differential equations for the pentagon functions, which enable efficient numerical evaluation via generalised power series expansion method."

to

"We derive differential equations for the pentagon functions and solve them numerically using the generalised power series expansion method. "

6) We modify the last sentence of the second-to-last paragraph in the introduction to specify explicitly that we work with on-shell top quarks. The sentence reads

"Furthermore, we consider the top quark on-shell and work in the helicity-amplitude representation, which provides a framework ... "

7) We have clarified the inconsistency by making the change from

"Black solid lines represent top quarks while black dashed lines represent massless external quarks."

to

"Black solid (dashed) lines represent top (massless) quarks associated with open fermion lines."

8) We have explicitly state the values of n_f and n_h used in the numerical benchmark evaluation in Eq.(69) of the updated version of the manuscript.

9) We have added the following just below Eq.(25), the second sentence:

"This set of reference vectors is chosen arbitrarily, however, different choice of such vectors will lead to different set of tensor structures."

10) We change the sentence just after Eq.(25) as follows:

"In writing $\Gamma_{\mathrm{g},i}$ and $\Gamma_{\mathrm{q},i}$, we have chosen $p_4$, $p_3$ and $p_3$ as the reference momenta of the polarisation vectors $\vareps(p_3)$, $\vareps(p_4)$ and $\vareps(p_5)$, respectively."

-->

"In writing $\Gamma_{\mathrm{g},i}$ and $\Gamma_{\mathrm{q},i}$, $\vareps(p_i)\cdot p_i = 0$ and $\vareps(p_i)\cdot q_i = 0$ (where $q_i$ is the reference momentum associated with the polarisation vector $\vareps(p_i)$) conditions have been enforced and we have chosen $q_3=p_4$, $q_4=p_3$ and $q_5=p_3$."

11) We have modified the following sentence and change the curly bracket to parentheses:

" ... are listed in table 1 for the standard permutation, σ0 = {1, 2, 3, 4, 5}. "

to

" ... are listed in table 2 for the standard permutation σ0 = (1, 2, 3, 4, 5), where the standard permutation refers to the ordering of the external momenta shown in fig. 2. "

12) We add the following sentence after Eq.(49) to introduce the notion of "transcendental weight":

"In this case, the order of the expansion ω also represents the transcendental weight, which counts the number of iterated integrations."

13) We change the following sentence at the last paragraph of Section 3.1:

"For constructing the alphabet across all the families and their permutations, we collected all the letters and look for linear relations between them in order to identify a minimal set."

-->

"To construct the alphabet across all families and their permutations, we start from the set of letters associated with the standard permutation of each family, as derived in [77]. We then generate all corresponding sets under the kinematic permutations listed in eq. (44) and take their union obtaining an over-complete alphabet. We then identify linear relations among the dlogs of the resulting letters to determine a minimal set of letters."

14) We specified that the relations are among dlogs of the letters, see answer to point 13.

15) We specify in the text that we mean “algebraically” independent components.

16) We changed appropriately the text so that it is clear that the pentagon functions basis leads to the cancelation of poles:

"The latter leads to significant simplifications in the final expression of the amplitude, to the analytic subtraction of ε-poles, and improves the efficiency of the numerical evaluation."

17) We added in the text: "For the latter, in order to determine the numerical boundary values of the MIs, we evaluate them at an arbitrary point of the physical phase space, $\vec{x_0}$, using \textsc{AMFlow}~\cite{Liu:2022chg} interfaced to \textsc{LiteRed} and \textsc{FiniteFlow} for the IBP reduction. The \textsc{AMFlow} package utilizes the auxiliary mass flow method~\cite{Liu:2017jxz} for the numerical evaluation of MIs. We evaluate the MIs at $\vec{x_0}$ asking for $70$-digit precision. This level of precision exceeds what is expected to be necessary for future applications, but it is employed here since evaluating the MIs at a single point with AMFlow does not constitute a computational bottleneck."

18) In section 3.4 we specify:

"These points are chosen to lie in the $s_{34}$ channel, defined in section \cite{sec:s34channel}, and are randomly generated with the only constraint being to avoid spurious singularities in the DEs."

and

"The boundary values for the set of pentagon functions $\vec{G}(\vec{x})$ are determined using numerical evaluations of the MIs performed with \textsc{AMFlow}, requesting a precision of 70 digits. "

Also in section 3.4 we add:

Moreover, in section 4 we add: "In the \textsc{Mathematica} notebook that we provide to evaluate the squared matrix elements for both $\ttj$ and $\ttgamma$ production, the numerical entries for the momenta are given to 100-digit precision, while the numerical values of the pentagon function basis have been computed with LINE to an accuracy of 32 digits."

19) We define $[]_{\vec{x}_0} (\vec{x})=1$ for $\omega=0$ after eq. (49).

20) We make this explicit before eq. (48):

"Then, we iteratively integrate the canonical DEs order by order in $\eps$ along a path $\gamma: [0,1] \to \gamma (t)$ connecting the initial point $\vec{x}_0$ to an arbitrary point of the phase space $\vec{x}$."

21) We replaced Z4 in favor of (2/5) Z2^2 in our final expressions. We included iπ in our ansatz for building the pentagon functions as a check, and verified that doesn't appear in our final expressions as expected. For avoiding confusions we modified appropriarly the text of the article above and under eq. (54).

22) We make this more clear in the text by adding:

"More specifically, the right-hand side of \cref{eq:beyond_symbol} is set equal to the right-hand side of \cref{Chen_MIs} at weight two, where the pentagon functions are expressed in terms of MI components and subsequently in terms of Chen iterated integrals."

23) We modify the following sentence in the second-to-last paragraph in Section 3.2 from

"Given that the boundary values are already computed using AMFlow, we can determine the coefficients $\tilde{c}$ and $\tilde{c}^i$."

to

"Given that the boundary values are already computed using AMFLOW, we can determine the coefficient $\tilde{c}$ only numerically, and then we rationilize their values. To verify the correctness of this result, we evaluate the left-hand side of eq. (54), including the contribution of the $\tilde{c} \zeta_2$ term, at a random point in the physical phase space using AMFLOW to obtain the values of the pentagon functions. We then compare this result with the numerical evaluation of the corresponding MI components on the right-hand side of eq. (54), also performed with AMFLOW. This comparison is carried out at five different phase-space points."

24) We add a sentence below Eq.(61) to explain how a square-root free connection matrix is achieved:

"In particular, we construct N so as to multiply the pentagon functions that are associated with MIs containing square roots in their normalization by the corresponding square roots."

25) We refer to the DEs of each family separately. We rephrased the text appropriately:

"The system of DEs for the MIs associated with one of the families in φ at a given permutation is significantly smaller, since it involves only a single permutation, whereas the pentagon functions cover the full set of permutations and families. However, for a given phase-space point, the numerical solutions for the MIs have to be computed multiple times (once for each permutation), while in the pentagon-function representation they are evaluated only once."

26) We have explicitly give the values of n1 and n2 reference vectors entering the helicity amplitudes in the numerical evaluation in Eq.(70) of the updated manuscript.

---

## Round 1 · Referee Report · Anonymous (Referee 2) · 2025-10-29

Report

The work is dedicated to calculation of analytic expressions for the full-colour one-loop helicity amplitudes for two processes ttj and ttgamma up to O(epsilon2). This work is timely and important and it will be employed for the calculation of 2 -> 3 two-loop scattering amplitudes. For ttgamma production the results provided for the one-loop scattering amplitude are novel. In the case of ttj, the new part involves deriving the pentagon function basis for this process. In both cases using the pentagon-function basis is very well motivated and leads to a significant simplification of the final expressions. This would be of great importance when calculating the NNLO QCD corrections for both processes.

The paper fulfils the necessary criteria to permit a publication in SciPost. Before
recommending the publication, however, I would like the authors to
address my comments below.

Requested changes

On page 14 the authors write that in order to determine the numerical boundary values of the MIs, they evaluate them at an arbitrary point of the physical phase space using AMFLOW with 70-digit precision.

Could the authors explain why such precision is needed? Wouldn't 34-digits precision, for example, be sufficient? What is the final precision the authors are aiming for here?

Small typos I found: - page 3 after citations [72–76] of a calculation --> for a calculation - Eq (54) a full stop at the end of a sentence is missing - Eq (55) should be a full stop instead of a comma - Eq (57) should be a comma instead of a full stop

I would suggest that you check all punctuation marks in the paper.

Recommendation

Ask for minor revision

  • validity: high
  • significance: high
  • originality: high
  • clarity: high
  • formatting: excellent
  • grammar: excellent

Author:  Dhimiter Canko  on 2025-11-14  [id 6030]

(in reply to Report 2 on 2025-10-29)

We would like to thank both referees for their thoughtful and constructive comments and suggestions. We believe their feedback has been very helpful and has significantly improved the quality of our paper. Here we answer the comments and questions raised by Referee #2.

We have addressed all the points raised in the report as detailed below.

  1. The referee asked for clarification regarding the choice of 70-digit precision in the evaluation of the master integrals using AMFlow, and whether a lower precision (e.g., 34 digits) would be sufficient. We have now added an explanatory sentence in the revised version of the manuscript: “This level of precision exceeds what is expected to be necessary for future applications, but it is employed here since evaluating the MIs at a single point with AMFlow does not constitute a computational bottleneck.”

  2. We thank the referee for carefully pointing out several minor typographical and punctuation errors. We have corrected all of the mentioned issues.

Anonymous on 2025-11-14  [id 6033]

(in reply to Dhimiter Canko on 2025-11-14 [id 6030])

We have uploaded in the attachments the new version of the manuscript.

Attachment:

tta_ttj_1l_SciPost_v2.pdf

---

## Round 1 · Referee Report · Anonymous (Referee 2) · 2025-11-17

Report

The authors satisfactorily addressed all questions raised. I recommend
publication in SciPost.

Recommendation

Publish (easily meets expectations and criteria for this Journal; among top 50%)

---

## Round 1 · Referee Report · Anonymous (Referee 1) · 2025-11-21

Report

I thank the authors for the clarifications. They have fully addressed my feedback in the second version of their manuscript, and I am now happy to recommend its publication in SciPost.

Recommendation

Publish (easily meets expectations and criteria for this Journal; among top 50%)

---

## Editorial Decision

published